# RETHINKING CODE SIMILARITY FOR AUTOMATED ALGORITHM DESIGN WITH LLMS

**Rui Zhang**
Department of Computer Science
City University of Hong Kong
`rui.zhang.cs@my.cityu.edu.hk`

**Zhichao Lu**[*]
Department of Computer Science
City University of Hong Kong
`zhichao.lu@cityu.edu.hk`

## ABSTRACT

The rise of Large Language Model-based Automated Algorithm Design (LLM-AAD) has transformed algorithm development by autonomously generating code implementations of expert-level algorithms. Unlike traditional expert-driven algorithm development, in the LLM-AAD paradigm, the main design principle behind an algorithm is often implicitly embedded in the generated code. Therefore, assessing algorithmic similarity directly from code, distinguishing genuine algorithmic innovation from mere syntactic variation, becomes essential. While various code similarity metrics exist, they fail to capture algorithmic similarity, as they focus on surface-level syntax or output equivalence rather than the underlying algorithmic logic.

We propose BehaveSim, a novel method to measure algorithmic similarity through the lens of problem-solving behavior as a sequence of intermediate solutions produced during execution, dubbed as problem-solving trajectories (PSTrajs). By quantifying the alignment between PSTrajs using dynamic time warping (DTW), BehaveSim distinguishes algorithms with divergent logic despite syntactic or output-level similarities. We demonstrate its utility in two key applications: (i) Enhancing LLM-AAD: Integrating BehaveSim into existing LLM-AAD frameworks (e.g., FunSearch, EoH) promotes behavioral diversity, significantly improving performance on three AAD tasks. (ii) Algorithm analysis: BehaveSim clusters generated algorithms by behavior, enabling systematic analysis of problem-solving strategies—a crucial tool for the growing ecosystem of AI-generated algorithms. Data and code of this work are open-sourced at `https://github.com/RayZhhh/behavesim`.

## 1 INTRODUCTION

The emerging paradigm of Large Language Model-based Automated Algorithm Design (LLM-AAD) (Liu et al., 2026) has garnered significant interest for its potential to autonomously generate code implementations of expert-level algorithms. This approach integrates a LLM into an iterative search framework (e.g., an evolutionary algorithm), where the LLM proposes candidate algorithms and the search routine governs the overall process. Crucially, LLM-AAD inverts the traditional, expert-driven workflow: instead of first articulating an algorithm's design logic and then implementing it in code, the core ideas are often implicitly encoded in the generated algorithm implementation. Consequently, as LLM-AAD sees broader applications, the ability to assess algorithmic similarity directly from code—distinguishing genuine innovation from mere syntactic variation—becomes essential.

Measuring similarity between code snippets has been extensively studied in software engineering, with widespread applications in diverse tasks such as code retrieval (Keivanloo et al., 2014), clone detection (Roy et al., 2009), and program classification (Mou et al., 2016). Existing code similarity measurements can be broadly classified into two main categories. Methods in the first category analyze programs in a static setting without executing them, i.e., calculating similarity based on features derived from their code. Some representative features used by methods in this category

---

[*]Corresponding Author

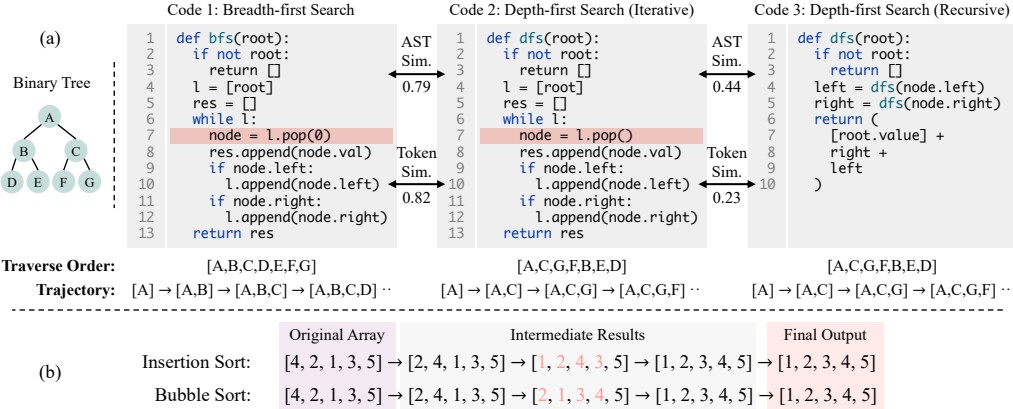

Figure 1: Examples demonstrating existing code similarity metrics are insufficient for measuring algorithmic similarity. **(a)** Existing code similarity metrics, on the one hand, find the breadth-first search (BFS) and depth-first search (DFS) algorithms highly similar, despite the two algorithms being inherently different in their traversal logic. On the other hand, they also find that a re-implementation of DFS (`Code 3`) based on recursion is a completely different algorithm from the original iteration-based implementation (`Code 2`), despite the two codes essentially representing the same algorithm. **(b)** Merely checking the output of two algorithms cannot distinguish between two distinct algorithms. Both insertion sort and bubble sort algorithms yield identical sorted arrays, despite being inherently different algorithms.

include token-based (Papineni et al., 2002), structure-based (Ren et al., 2020), and embedding-based (Dong et al., 2025). In contrast, methods in the other category assess similarity by executing the codes and comparing the generated outputs over a suite of test cases (Roziere et al., 2020).

However, a fundamental limitation arises when applying existing code similarity metrics to assess algorithmic similarity: *they primarily evaluate surface-level syntax or output equivalence rather than capturing the underlying algorithmic logic*. For example, in the binary tree traversal problem (Figure 1(a)), existing static feature-based metrics cannot differentiate between algorithms with syntactically similar implementations but divergent underlying logic. Similarly, existing execution-based metrics (Figure 1(b)) conflate algorithms like insertion sort and bubble sort, as they produce identical outputs despite differing in the sorting logic. These cases collectively demonstrate that accurate assessment of algorithmic similarity must account for an algorithm's problem-solving behavior — specifically, the approach it employs to solve a given task.

In this work, we introduce **BehaveSim**, a tangible method for measuring similarity between algorithms from the behavioral perspective. Specifically, we propose to define behavior similarity based on the problem-solving trajectory (PSTraj) of an algorithm, where a PSTraj is a sequence of intermediate or partial solutions generated by an algorithm on a problem. Then, we measure the differences between two PSTrajs by aggregating over the pairwise distances between elements from these two PSTrajs via dynamic time warping (DTW) (Senin, 2008). A pictorial illustration is provided in Figure 2, showing that BehaveSim can differentiate "lookalike" algorithms with distinct problem-solving behaviors. In essence, BehaveSim provides a new angle to quantify the novelty of an algorithm from the perspective of behavior similarity (to existing algorithms).

Furthermore, we demonstrate two direct use cases (UCs) of BehaveSim:

[UC1] *Improving the performance of existing LLM-AAD methods*: Recent advances have suggested that maintaining diversity among candidate algorithms is crucial for guiding the search towards high-quality algorithms (Romera-Paredes et al., 2024; Wang et al., 2024a). BehaveSim provides a new way to control diversity by encouraging algorithms with distinct problem-solving behaviors. Empirically, we demonstrate that both FunSearch (Romera-Paredes et al., 2024) and EoH (Liu et al., 2024) with BehaveSim significantly outperform their original counterparts and other existing state-of-the-art methods on three AAD tasks.

[UC2] *A new tool for quantitative algorithm analysis*: As LLM-AAD is gaining popularity, an increasing number of AI-generated algorithms are expected in the near future. Being able to

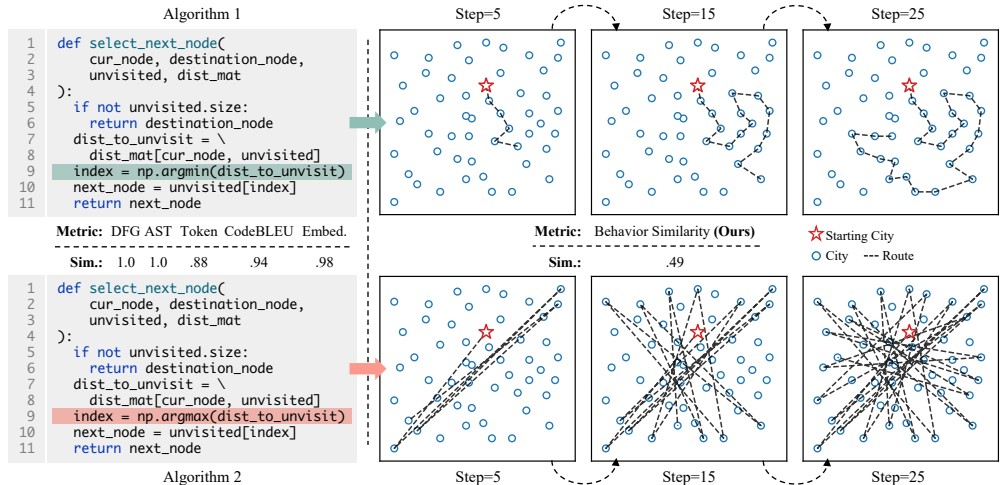

Figure 2: Problem-solving behaviors on the traveling salesman problem (TSP) for two algorithms with highly similar codes. The only distinction in their implementations lies in the use of `argmin()` and `argmax()`, which leads to profoundly different behaviors: Algorithm 1 chooses the nearest neighbor node of the current node, while Algorithm 2 always steps to the farthest node away from the current node. Nevertheless, existing similarity metrics still assign them a high degree of similarity, failing to reveal their behavioral differences.

automatically and quantitatively analyze algorithms becomes critically important to ensure the sustainable development of LLM-AAD. Taking algorithms generated by an existing LLM-AAD method as an example, we demonstrate that BehaveSim can organize generated algorithms into clusters with similar behavior, facilitating the discovery and analysis of distinct problem-solving strategies.

In summary, our primary contributions are as follows.

1. We demonstrate the necessity of measuring algorithmic similarity through the perspective of their problem-solving behaviors.

2. We propose BehaveSim, a tangible method for measuring behavioral similarity based on the problem-solving trajectory. We demonstrate its significance and effectiveness from both methodological and empirical perspectives.

3. We demonstrate the effectiveness of BehaveSim in two use cases: (i) enhancing existing LLM-AAD methods by promoting behavioral diversity during search, and (ii) providing a quantitative tool for analyzing algorithms based on their problem-solving behaviors.

## 2 REVISITING CODE SIMILARITY

This section provides an overview of existing code similarity metrics and an empirical study evaluating their effectiveness in measuring behavioral similarity.

### 2.1 EXISTING CODE SIMILARITY METHODS

A common approach in existing code similarity methods first derives features from code implementations and then measures similarity based on those features.

**Token-based** methods treat source code as natural language text. Methods such as BLEU (Papineni et al., 2002) and ROUGE (Lin, 2004) parse the source code into a sequence of tokens and match their similarity using N-Gram. In particular, CrystalBLEU (Eghbali & Pradel, 2022) and CodeBLEU (Ren et al., 2020) improve N-Gram by emphasizing programming language-specific tokens.

**Structure- and Graph-based** methods calculate the similarity between higher-level representations of code implementations. These include abstract syntax tree (AST) (Gabel et al., 2008; Ren et al.,

2020), data flow graphs (DFG) (Ren et al., 2020), control flow graphs (Zhao & Huang, 2018), and program dependency graphs (PDG) (Liu et al., 2023).

**Embedding-based** methods represent code in a learned vector space (Maveli et al., 2025; Günther et al., 2023). These approaches typically employ pre-trained models to capture semantic similarities or to compare token-level embeddings (Maveli et al., 2025; Zhou et al., 2023; Zhang et al., 2020). For example, CodeBERTScore (Zhou et al., 2023) computes the cosine similarity between the embeddings encoded by a fine-tuned CodeBERT (Feng et al., 2020) and calculates the similarity based on the $F_1$ score of the best matching token pairs.

**Execution trace-based** methods measure the similarity between low-level execution traces (Pei et al., 2020), which typically comprise dynamics of variable values at executed lines (Ni et al., 2024), fine-grained program-state changes at the instruction level (e.g., x86 instructions) (Pei et al., 2020), or variable attributes (e.g., addresses and sizes) (Wang et al., 2024b).

## 2.2 LIMITATION OF EXISTING METRICS FOR ALGORITHMIC SIMILARITY

**A Methodological Perspective.** While existing static code similarity methods (i.e., token-, structure-, graph-, and embedding-based methods) are effective at software engineering tasks (e.g., clone detection, code retrieval), they struggle to capture the dynamic problem-solving behavior of algorithms. As illustrated in Figure 2, the only difference between Algorithm 1 and Algorithm 2 lies in the minor modification of a function (`argmin()` vs. `argmax()`), yet this leads to a substantial discrepancy in their behavior: Algorithm 1 controls the step toward the nearest neighbor node, while Algorithm 2 always chooses the farthest node. Despite significant differences in their behavior, existing code similarity metrics yield a high similarity score. Notably, both their DFG and AST similarities are 1.0, indicating identical structural features in their code. This reveals that they likewise struggle to measure the underlying behavior of algorithms.

On the other hand, execution-trace-based methods (Pei et al., 2020; Ni et al., 2024; Wang et al., 2024b) track the dynamics for multiple variables and function calls, which contain rich and useful information for general-purpose program analysis. However, when characterizing problem-solving behavior, the large number of low-level state changes may introduce excessive information unrelated to the algorithm's problem-solving dynamics, potentially obscuring similarity calculations. In this regard, BehaveSim can be viewed as a specialized, focused execution trace of intermediate solutions, rather than an exhaustive record of all program variables' state changes.

**An Empirical Perspective.** In addition, we curate a dataset to systematically assess whether existing code similarity metrics effectively capture algorithmic similarity. Algorithmic similarity can be characterized along three dimensions: (1) textual appearance, (2) outputs/results, and (3) procedural behavior (i.e., the steps taken to derive results). This yields eight possible similarity types, as detailed in Table 1. We exclude trivial cases in which all three dimensions are either entirely similar or entirely dissimilar, as well as infeasible scenarios in which a behaviorally similar algorithm produces divergent outputs. Consequently, our dataset construction focuses on the four remaining non-trivial types (Type-1 to Type-4).

Accordingly, our dataset consists of four distinct types of algorithm pairs, designed to disentangle similarities in text, results, and behavior. Below, we provide brief explanations and representative examples for each type. Detailed specifications for individual data pairs are presented in Appx. § A.

Table 1: Eight possible types of algorithmic similarity. "✓" indicates similarity in a given dimension (textual appearance, results, or behavior), while "✗" indicates dissimilarity. After excluding trivial cases, the dataset focuses on the four non-redundant types (Types 1–4).

| | Text | Result | **Behavior** |
|---|---|---|---|
| Type-1 | ✓ | ✓ | ✗ |
| Type-2 | ✓ | ✗ | ✗ |
| Type-3 | ✗ | ✓ | ✓ |
| Type-4 | ✗ | ✓ | ✗ |
| $\binom{\text{Not}}{\text{Interested}}$ | ✓ | ✓ | ✓ |
| | ✗ | ✗ | ✗ |
| $\binom{\text{Not}}{\text{Feasible}}$ | ✓ | ✗ | ✓ |
| | ✗ | ✗ | ✓ |

- Type 1: High text similarity, different behaviors, similar results. These pairs consist of algorithms that are textually similar but exhibit different behaviors. A typical example is matrix multiplication, where subtle differences in the order of `i`, `j`, `k` lead to divergent internal computations, despite a high degree of code-level overlap.

Table 2: Average similarity on four types of data calculated by various metrics.

| Method Type | Method Name | Type-1 | Type-2 | Type-3 | Type-4 |
|---|---|---|---|---|---|
| Based on Token | ROUGE | 0.95 | 0.96 | 0.70 | 0.47 |
| | BLEU | 0.83 | 0.94 | 0.42 | 0.16 |
| | CrystalBLEU | 0.97 | 0.99 | 0.68 | 0.51 |
| Based on Structure | AST | 0.96 | 1.00 | 0.76 | 0.57 |
| Combine Token and Structure | CodeBLEU | 0.97 | 0.94 | 0.91 | 0.75 |
| Based on Embedding | CodeBertScore | 0.84 | 0.97 | 0.60 | 0.38 |
| | Jina-Code-Embedding | 0.99 | 0.99 | 0.90 | 0.84 |
| | Qwen3-Embedding-0.6B | 0.94 | 0.93 | 0.87 | 0.73 |
| Based on Execution Results | – | 1.00 | 0.00 | 1.00 | 1.00 |
| Based on Execution Trace | Exe-Trace+BLEU | 0.86 | 0.95 | 0.61 | 0.54 |
| | Exe-Trace+Jina-Code-Emb | 1.00 | 1.00 | 0.87 | 0.77 |
| | Exe-Trace+Qwen3-Emb-0.6B | 0.99 | 1.00 | 0.91 | 0.78 |
| Based on behavior (Ours) | BehaveSim | 0.56 | 0.73 | 1.00 | 0.46 |

- Type 2: High text similarity, different behaviors and results. These algorithms are similar in text, but a small, critical change (e.g., a parameter or a conditional statement) not only alters their behavior but also yields different final outputs.

- Type 3: Low text similarity, similar behaviors and results. This category contains algorithms implemented in diverse ways (e.g., an iterative vs. a recursive implementation of Depth-First Search) but that produce the same behavior and results.

- Type 4: Low text similarity, different behaviors, similar results. Algorithms of this type are distinct not only in their code implementation but also in their behavior. However, they produce the same final output. For example, both quicksort and bubblesort yield a sorted array, but their behavior when swapping each subarray is completely different.

We evaluate six categories of code similarity metrics on the curated dataset: ❶ Token-level metrics, including ROUGE, BLEU, and CrystalBLEU; ❷ Structure-level metrics, such as AST similarity; ❸ Hybrid metrics, such as CodeBLEU, which combines token and structural features via a weighted average; ❹ Embedding-based metrics, including CodeBERTScore (Zhou et al., 2023) and cosine similarity of two embedding models, Jina-Code-Embedding (Günther et al., 2023) and Qwen3-Embedding-0.6B (Zhang et al., 2025); ❺ Execution-based metrics, which evaluate whether two algorithms yield identical results on the same set of inputs. ❻ Execution-trace-based methods. We trace the internal variable dynamics of algorithms using `pysnooper`, converting the traces into tokens and embeddings, and measure their similarity via N-Gram methods (e.g., BLEU) and embedding models (Jina-Code-Embedding, Qwen3-Embedding-0.6B).

We report the average similarity value on each type of data in Table 2, with results on individual data pairs within each type listed in Appx. § A. We observe from the results that:

- Both static similarity methods (token-, structure-, and embedding-based methods) and execution-trace-based methods consistently yield high similarity for Type-1 and Type-2 data, which have similar code. However, they fail to measure Type-3 pairs, which have similar behavior but dissimilar code, indicating they cannot distinguish algorithmic behavior.

- The Execution-based method correctly identifies Type-3 and Type-4 pairs as having identical results. It cannot distinguish between Type-1 and Type-4 pairs, as both produce identical final outputs despite exhibiting different behaviors, suggesting that simply checking outputs is insufficient for evaluating behavioral similarity.

- Although execution-trace-based methods incorporate dynamic execution information, they still exhibit high similarity on Type-1 and Type-2 pairs, as execution traces may contain excessive noise information (e.g., variable state changes) that obscures the underlying problem-solving logic.

In summary, the results demonstrate that existing algorithmic similarity metrics consistently fail to distinguish between problem-solving behaviors.

# 3  BEHAVESIM: MEASURING BEHAVIORAL SIMILARITY BETWEEN ALGORITHMS

**Scope of Algorithms.**   This work focuses on a general class of algorithms that progressively refine solutions through successive updates, as opposed to one-shot solution generation. Formally, we define an iterative algorithm as a mapping $f : \mathbf{x} \to \mathbf{x}$ where $x_{t+1} = f(x_t)$ with $x_t$ representing the solution at iteration $t$. This formulation encompasses a broad spectrum of important algorithms, including search and sorting algorithms, optimization methods (e.g., gradient descent, heuristics), and many machine learning training procedures.

**Problem-solving Trajectory (PSTraj).**   To measure behavioral similarity between algorithms, it is essential to first establish a quantitative representation of problem-solving behavior. We represent this behavior as a problem-solving trajectory (PSTraj) on a problem instance. Specifically, a PSTraj consists of the intermediate solutions generated by an iterative algorithm during problem-solving. Iterative algorithms start from an initial solution $x_0$ and progressively produce a solution or partial solution at each step. For example, in convex optimization, an intermediate solution may be a complete but suboptimal vector; in binary tree traversal, a partial solution might refer to a subset of nodes that does not yet constitute a full path. We collect these progressively generated intermediate solutions into a sequence $\mathcal{T} = (x_0, x_1, \ldots, x_T)$, where $T$ denotes the total number of iterations. We refer to this sequence as the algorithm's PSTraj. This sequence provides a tangible quantification for characterizing an algorithm's problem-solving behavior.

**Pairwise Distance Between Two Solutions.**   We first define a distance measure between individual solutions before measuring the distance between two PSTraj. The calculation of pairwise distance depends on the solution types of the problem. The pairwise distance between different types of solutions is defined as follows.

- Categorical, ordinal, permutation solutions. For these types, we adopt the edit distance $d_{\mathrm{edit}}(x, y)$ as the pairwise distance. To ensure comparability across problem instances, the distance is normalized as: $d(x, y) = d_{\mathrm{edit}}(x, y)/d_{\max}$ where $d_{\max}$ is the maximum possible edit distance for the given problem.

- Discrete and continuous solutions. For these cases, we employ the Euclidean distance $d_{\mathrm{euc}}(x, y) = \|x - y\|_2$, normalized by a problem-specific upper bound of distance $D$ (e.g., the possible maximum distance in the domain): $d(x, y) = \|x - y\|_2/D$.

**Similarity Between PSTrajs.**   Given the pairwise distances between solutions, we define the distance between two trajectories using the Dynamic Time Warping (DTW) (Senin, 2008). DTW is particularly suitable because it aligns trajectories of potentially different lengths or with temporal shifts, enabling it to capture local similarities of two trajectories. Formally, given two trajectories $X = (x_1, \ldots, x_m)$ and $Y = (y_1, \ldots, y_n)$, the DTW distance is defined as:

$$DTW(X, Y) = \min_{\pi \in \mathcal{A}(m,n)} \sum_{(i,j) \in \pi} d(x_i, y_j),$$

where $\mathcal{A}(m, n)$ denotes the set of all possible alignment paths between the two trajectories. We define $\mathrm{Sim}_{\mathrm{PSTraj}}$ by normalizing the DTW distance by the length of the shorter trajectory:

$$\mathrm{Sim}_{\mathrm{PSTraj}}(X, Y) = 1 - \frac{DTW(X, Y)}{min\{|X|, |Y|\}},$$

We note that alternative methods for calculating trajectory distance are also feasible in this scenario. For example, we may calculate the mean pairwise distance for respective solutions. Further discussion on the choices of trajectory distances is provided in Appx. §B.

**BehaveSim.**   Given a target problem (e.g., traveling salesman problem) and $P^I$, a probability distribution of its problem instances, we let:

- $I \sim P^I$ be the problem instance sampled from the distribution $P^I$,
- $s \sim S_I$ be the starting point (or initial solution) sampled from $S_I$, which is the distribution of feasible starting points on a specific problem instance $I$,

- $\mathcal{T}(A, I, s)$ be the PSTraj generated by algorithm $A$ on instance $I$ from start-point $s$,
- $\text{Sim}_{\text{PSTraj}}(\mathcal{T}_1, \mathcal{T}_2)$ be the similarity score between two PSTrajs.

We define the behavioral similarity BehaveSim between two algorithms, $A_1$ and $A_2$, as the expected similarity of their problem-solving trajectories:

$$\text{BehaveSim}(A_1, A_2) := \mathbb{E}_{I \sim P^I} \left[ \mathbb{E}_{s \sim S_I} \left[ \text{Sim}_{\text{PSTraj}} \left( \mathcal{T}(A_1, I, s), \mathcal{T}(A_2, I, s) \right) \right] \right]$$

Critically, this expectation is calculated over two nested levels of variation: (i) the distribution of problem instances $I$ for a given problem distribution $P^I$, and (ii) for each instance, the distribution of possible starting points $S_I$. Moreover, we highlight two additional considerations in the practical utilities of our method:

- Processing PSTraj via truncation or sampling. It is sufficient to select a subset of PSTraj to represent the algorithms' behavior. Shorter PSTraj can save computational cost for BehaveSim calculation. BehaveSim controls trajectory granularity via ❶ Truncation (removing the last proportion $k$ of the PSTraj); or ❷ Sampling (keeping one solution every $n$ steps). We analyze the influence of these parameters in Appx. §C.

- BehaveSim for algorithms with stochastic internal state. Stochastic intermediate states do not invalidate the concept of behavioral similarity, but we need new ways to measure the pairwise distance. ❶ If the transition $x_{t+1} = f_A(x_t \mid \xi)$ involves a random noise $\xi$, we obtain multiple samples to approximate the distribution $P_{t+1}(x)$ and similarly obtain $P_{t+1}(y)$ for algorithm $B$. The pairwise distance can then be defined as a divergence $D(P_{t+1}(x), P_{t+1}(y))$. ❷ Fixing the random seed is another way to reduce the randomness. Sampling multiple solutions at each step is more robust because it compares behavioral distributions, but it incurs greater computational overhead. Fixing random seeds ensures reproducibility, but may lead to biased calculations of behavioral similarity. Depending on the specific context, either approach can be selected accordingly.

### 3.1 VISUALIZATION OF BEHAVESIM

First, we consider a continuous optimization problem of minimizing a two-variable Rosenbrock function, where each solution is a vector of two real numbers. Figure 3(a) visualizes the problem-solving trajectories of three different optimization algorithms, namely SGD, BFGS, and Conjugate Gradient (CG), on this problem. For each pair of algorithms, we report the average similarity score across two random initial starting points. As shown in the left two plots (SGD vs. BFGS), the trajectories are less similar, leading to lower similarity scores. In contrast, the right two plots (CG vs. BFGS) show more similar trajectories, leading to higher similarity scores.

Second, we consider a combinatorial optimization of the traveling salesman problem, where each solution (or partial solution) corresponds to a permutation of cities that defines the visiting order. As shown in Figure 3(b), we compare the problem-solving behavior of three expert-designed algorithms and visualize their partial solutions at steps 5, 25, and 50. The starting city is marked with a red star. We find that the similarity between Algorithm 1 and Algorithm 3 is higher than that between Algorithm 1 and Algorithm 2, as reflected in both the similarity results and intuitive observations, indicating that our approach can handle both continuous and permutation solution types and can reflect intuitive behavioral similarity between algorithms.

### 3.2 EMPIRICAL EVALUATION OF BEHAVESIM

We empirically evaluate BehaveSim on the code similarity benchmark dataset described in Sec. §2.2. As evidenced by the results in Table 2, BehaveSim accurately captures behavioral similarity, achieving a similarity score of 1.0 on Type-3 instances—correctly identifying algorithms with identical problem-solving logic despite syntactic differences. Furthermore, BehaveSim assigns significantly lower similarity values to Types 1, 2, and 4, demonstrating its robustness in distinguishing behavioral divergence. Notably, it remains unaffected by superficial textual overlap (Types 1 and 2) or output equivalence (Type 4), validating its focus on problem-solving dynamics rather than surface-level features. These results collectively underscore the effectiveness of BehaveSim for a quantitative evaluation of behavioral similarity in algorithm analysis.

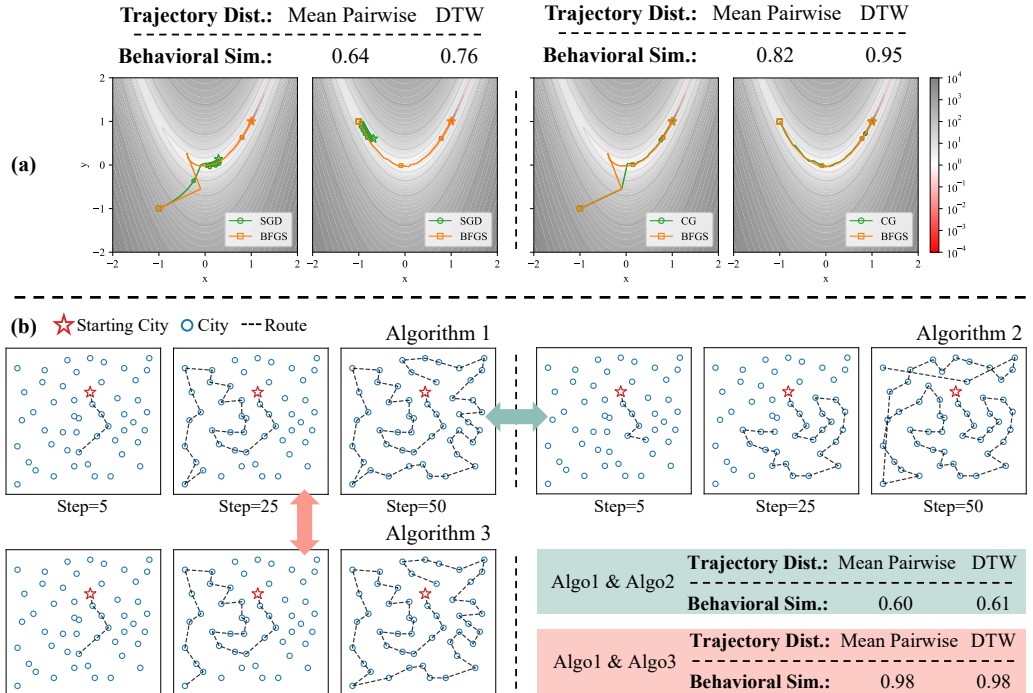

Figure 3: **(a)** Comparison of the problem-solving behavior for optimization algorithms. Each plot shows the trajectories of two algorithms on the Rosenbrock problem, each initialized at two different locations. Behavioral similarity is reported using both mean pairwise distance and DTW. **(b)** Comparison of TSP algorithms. Each plot shows the partial solution at different steps (5, 25, 50). The red star denotes the starting city.

## 4 Use Cases of BehaveSim

This section demonstrates two direct use cases of BehaveSim in LLM-AAD and algorithm analysis. BehaveSim enables direct control over the behavioral diversity of generated algorithms, potentially enhancing search performance. BehaveSim also provides a behavioral perspective for algorithm analysis, offering new insights beyond traditional code-level analysis.

### 4.1 Integration in LLM-AAD Method

**Method.** BehaveSim can be integrated into existing LLM-AAD methods to enhance the behavioral diversity of the candidate algorithms. A brief review of related work on LLM-AAD is provided in Appx.§D. In this work, we explore two such integration strategies and apply them to two representative LLM-AAD methods.

First, we propose a niche-based integration, which we demonstrate by combining with Fun-Search (Romera-Paredes et al., 2024). The key idea is to augment its multi-island database with a behavioral diversity management strategy, in which a new candidate algorithm is assigned to an island based on BehaveSim score rather than inheriting from its parent algorithm. We refer to this combination as FunSearch+BehaveSim.

Second, we use BehaveSim as a helper objective to augment the primary fitness score in EoH (Liu et al., 2024), in which we use the BehaveSim with the best algorithm found as the second objective during the search. We refer to this combination as EoH+BehaveSim. The implementation details of both methods are elaborated in Appx. §E.

**AAD Tasks.** We evaluate our approach on three AAD tasks, including Admissible Set Problem (ASP) (Romera-Paredes et al., 2024), Traveling Salesman Problem (TSP) (Matai et al., 2010), and Circle Packing Problem (CPP) (Novikov et al., 2025). We compare our search method against their

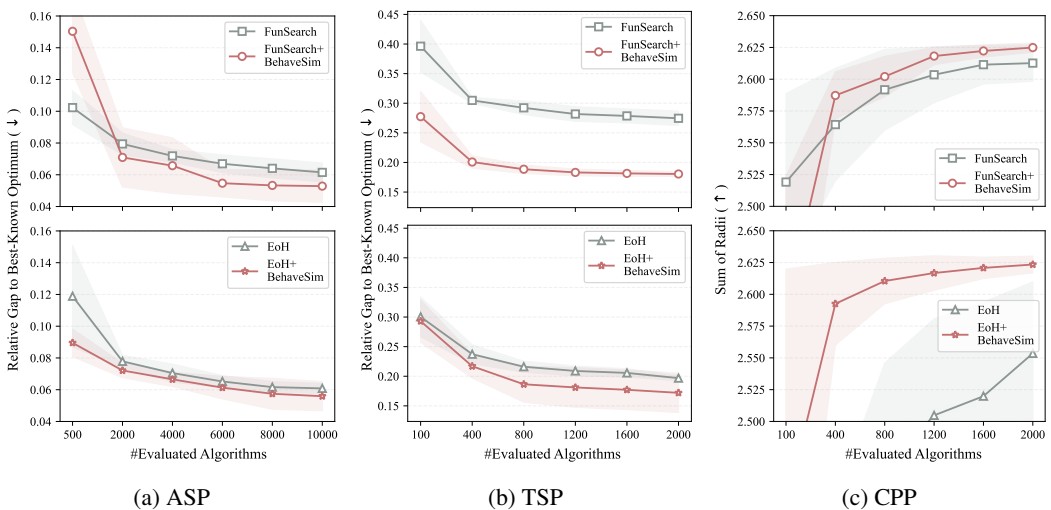

|     |     |     |
| :-: | :-: | :-: |
| (a) ASP | (b) TSP | (c) CPP |

Figure 4: Convergence of the top-10 algorithm sets produced by different search methods on three AAD tasks. The y-axis reports the relative gap to the best-known optimum (lower is better). Markers denote the mean over three independent runs; shaded regions indicate standard deviation.

original counterpart, FunSearch and EoH. The maximum number of evaluations is set to 2,000 for TSP and CPP, and 10,000 for ASP due to its higher complexity. Each candidate algorithm is evaluated with a 50-second timeout. All methods interact with the GPT-5-Nano[1] via API. More detailed settings on individual AAD tasks are provided in Appx. §F and Appx. §G.

Table 3: Performance of the top-1 and top-10 algorithms obtained by each AAD method. For ASP and TSP, metrics are the relative gap to the best-known optimum (%, **lower is better**). For CPP, the metric is the sum of radii (**higher is better**). All results are reported as mean $\pm$ standard deviation over three runs. Better results are highlighted in **bold**.

|  | ASP Performance ($\downarrow$) | | TSP Performance ($\downarrow$) | | CPP Performance ($\uparrow$) | |
| --- | :-: | :-: | :-: | :-: | :-: | :-: |
|  | Top-1 | Top-10 | Top-1 | Top-10 | Top-1 | Top-10 |
| FunSearch | $5.84_{\pm 0.55}$ | $6.15_{\pm 0.59}$ | $25.22_{\pm 0.56}$ | $27.46_{\pm 1.21}$ | $2.6259_{\pm 0.0051}$ | $2.6127_{\pm 0.0143}$ |
| FunSearch+BehaveSim | $\mathbf{5.24}_{\pm 1.05}$ | $\mathbf{5.28}_{\pm 1.01}$ | $\mathbf{17.37}_{\pm 0.23}$ | $\mathbf{18.05}_{\pm 0.42}$ | $\mathbf{2.6293}_{\pm 0.0026}$ | $\mathbf{2.6249}_{\pm 0.0036}$ |
| EoH | $5.99_{\pm 0.20}$ | $6.08_{\pm 0.31}$ | $18.86_{\pm 0.20}$ | $19.68_{\pm 0.42}$ | $2.5724_{\pm 0.0482}$ | $2.5535_{\pm 0.0562}$ |
| EoH+BehaveSim | $\mathbf{5.33}_{\pm 0.85}$ | $\mathbf{5.59}_{\pm 0.91}$ | $\mathbf{16.55}_{\pm 2.80}$ | $\mathbf{17.20}_{\pm 3.32}$ | $\mathbf{2.6263}_{\pm 0.0054}$ | $\mathbf{2.6234}_{\pm 0.0066}$ |

**Results.** Figure 4 presents the convergence curves of the performance of top-10 algorithms obtained by FunSearch+BehaveSim, EoH+BehaveSim, and their original counterparts. Complementary results for the top-1 and top-10 performances are reported in Table 3. The performance is calculated by the relative gap to the best-known optimum, with lower values indicating better performance.

We find that both FunSearch+BehaveSim and EoH+BehaveSim outperform their respective original counterparts, achieving the best results on both top-1 and top-10 algorithms across these tasks. These results indicate that coupling search methods with behavioral similarity improves both efficiency and final performance in LLM-AAD applications, underscoring the effectiveness of encouraging behavioral diversity. Due to the space limit, please refer to Appx. §F for ablation studies and analysis.

## 4.2 ALGORITHM ANALYSIS

BehaveSim offers a novel behavioral perspective for analyzing algorithms. To demonstrate this analytical utility, we evaluate 30 algorithms randomly sampled from the final database checkpoint of a FunSearch+BehaveSim run on the TSP. Figure 5 visualizes the hierarchical clustering dendrograms

---

[1]https://openai.com/index/introducing-gpt-5/

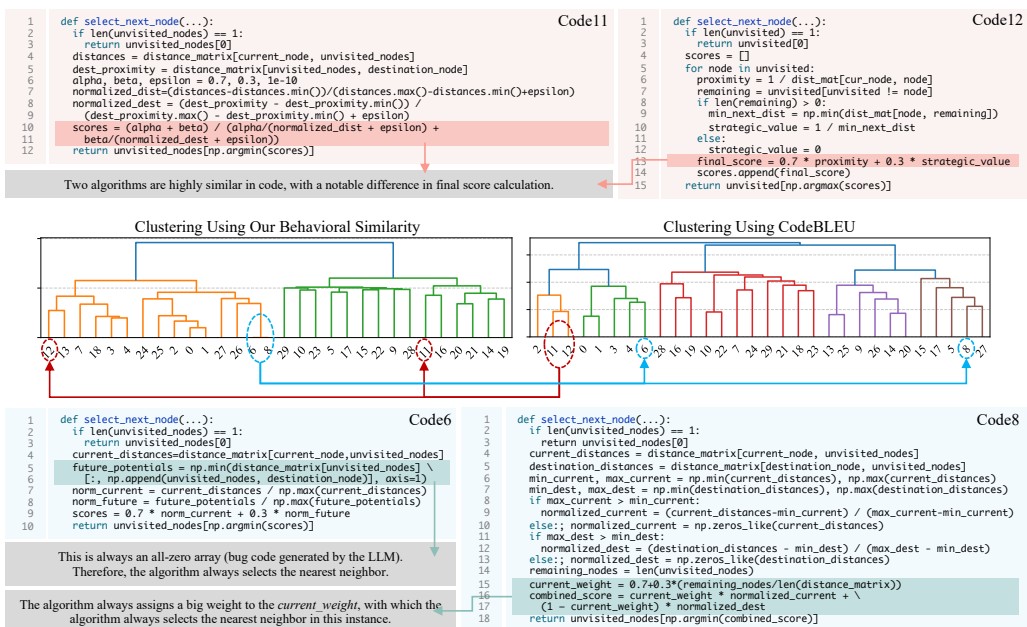

Figure 5: Clustering results based on two similarity measures. (i) Code 11 and Code 12 are clustered together by CodeBLEU, yet they exhibit distinct behaviors due to differences in their logic for computing the final score. (ii) Code 6 and Code 8 differ in code structure but display consistent behaviors. This is because a bug in Code 6 sets the "future_potentials" field to an all-zero array, making its score solely determined by the distance to neighbors. Similarly, Code 8 assigns a dominant weight to the distance term, which overrides other factors. As a result, both Code 6 and Code 8 essentially follow the strategy of selecting the nearest unvisited neighbor.

produced by BehaveSim and CodeBLEU. In these visualizations, a higher linkage distance (i.e., branching point) denotes a lower degree of similarity between the algorithms.

We observe a significant discrepancy in their clustering results. Notably, we find that Code 6 and Code 8 exhibit consistent behavior, as evidenced by their low-level merger in the left dendrogram (highlighted by the blue circle). However, they are distinct in code implementation, reflected by the significantly higher branching point between them in the right dendrogram. Qualitative analysis of Code 6 and Code 8 reveals that their behavioral alignment stems from a shared underlying logic: both inherently select the nearest node. Conversely, although Code 11 and Code 12 exhibit high structural similarity according to CodeBLEU (highlighted by the red circle in the right dendrogram), they diverge significantly in practice. This discrepancy originates from their distinct scoring mechanisms, which ultimately dictate their problem-solving behaviors.

These observations underscore the contributions of a behavioral perspective in algorithm analysis: it not only identifies the specific code segments responsible for critical behavioral shifts but also uncovers convergent design logic that might be obscured by implementation differences.

## 5 CONCLUSION

This paper introduces BehaveSim, a novel method measuring algorithmic similarity from a behavioral perspective. Through empirical comparisons with existing metrics, we conclude that BehaveSim can distinguish algorithms with disparate behaviors, whereas existing methods cannot. Two use cases of BehaveSim are demonstrated, showcasing the effectiveness of maintaining behavioral diversity in the algorithm search and its application in analyzing algorithm behaviors.

Moving forward, algorithmic similarity and novelty can be assessed along multiple dimensions beyond problem-solving behavior, such as time and space complexity. These perspectives are also crucial for understanding algorithmic properties. In this work, we focus solely on behavioral similarity, leaving other dimensions for future exploration.

ACKNOWLEDGMENTS

We sincerely thank Prof. Qingfu Zhang for his valuable feedback and constructive suggestions that significantly improved this work. We also gratefully acknowledge the LLM4AD team at the Optima Group, City University of Hong Kong, for insightful discussions and continuous support.

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

## A   DETAILED RESULTS ON ALGORITHMIC SIMILARITY DATASET

We provide similarity results for individual data pairs for each data type. Please refer to Table 4, Table 5, Table 6, Table 7, and Table 8 for detailed data pair specifications and results.

Table 4: Comparison of similarity metrics on **Type 1** cases, where code is syntactically similar, but execution paths are different while yielding similar final results. **Case 1**: Two matrix multiplication algorithms with different loop orders (ijk vs. jik). **Case 2**: Two BFS binary tree traversal algorithms, one expanding the left child first, the other expanding the right. **Case 3**: Two DFS binary tree traversal algorithms, one expanding the left child first, the other expanding the right.

| Method | Case 1 | Case 2 | Case 3 |
|---|---|---|---|
| ROUGE | 0.89 | 0.98 | 0.98 |
| BLEU | 0.65 | 0.93 | 0.92 |
| CrystalBLEU | 0.92 | 0.98 | 1.00 |
| AST | 0.88 | 1.00 | 1.00 |
| CodeBertScore | 0.61 | 0.96 | 0.96 |
| CodeEmbedding | 0.97 | 0.99 | 1.00 |
| CodeBLEU | 0.96 | 0.96 | 0.99 |
| **BehaveSim** | 0.32 | 0.69 | 0.68 |

Table 5: Comparison of similarity metrics on **Type 2** cases, where code is syntactically similar, but both execution paths and final results are different. **Case 1**: Two graph traversal algorithms differing only by `min()` vs. `max()` to select the next node (nearest vs. farthest). **Case 2**: Two online bin packing algorithms with slight hyperparameter variations. **Case 3**: Another pair of online bin packing algorithms with slight hyperparameter variations.

| Method | Case 1 | Case 2 | Case 3 |
|---|---|---|---|
| ROUGE | 0.95 | 0.97 | 0.97 |
| BLEU | 0.92 | 0.95 | 0.95 |
| CrystalBLEU | 0.99 | 0.99 | 0.99 |
| AST | 1.00 | 1.00 | 1.00 |
| CodeBertScore | 0.96 | 0.98 | 0.96 |
| CodeEmbedding | 0.99 | 1.00 | 0.99 |
| CodeBLEU | 0.88 | 0.97 | 0.97 |
| **BehaveSim** | 0.70 | 0.90 | 0.59 |

Table 6: Comparison of similarity metrics on **Type 3** cases, where code is syntactically dissimilar, but execution paths and final results are similar. **Case 1**: Recursive vs. iterative BFS. **Case 2**: Recursive vs. iterative Bubble Sort. **Case 3**: Recursive vs. iterative Insertion Sort. **Case 4**: Two implementations of Merge Sort. **Case 5**: Two equivalent implementations of First Fit for bin packing. **Case 6**: Two equivalent implementations of Best Fit for bin packing.

| Method | Case 1 | Case 2 | Case 3 | Case 4 | Case 5 | Case 6 |
|---|---|---|---|---|---|---|
| ROUGE | 0.74 | 0.61 | 0.57 | 0.51 | 0.88 | 0.91 |
| BLEU | 0.55 | 0.23 | 0.23 | 0.14 | 0.72 | 0.65 |
| CrystalBLEU | 0.75 | 0.57 | 0.58 | 0.62 | 0.88 | 0.70 |
| AST | 0.80 | 0.67 | 0.76 | 0.55 | 0.85 | 0.92 |
| CodeBertScore | 0.71 | 0.43 | 0.51 | 0.30 | 0.79 | 0.87 |
| CodeEmbedding | 0.91 | 0.88 | 0.89 | 0.82 | 0.96 | 0.95 |
| CodeBLEU | 0.85 | 0.87 | 0.88 | 0.89 | 0.98 | 0.99 |
| **BehaveSim** | 1.00 | 1.00 | 1.00 | 1.00 | 1.00 | 1.00 |

Table 7: Comparison of similarity metrics on **Type 4** cases where code and execution paths are dissimilar, but final results are similar (Part 1 of 2, Cases 1-9). Cases 1-14 involve pairwise comparisons of 6 different sorting algorithms.

| Method | 1 | 2 | 3 | 4 | 5 | 6 | 7 | 8 | 9 |
|---|---|---|---|---|---|---|---|---|---|
| ROUGE | 0.54 | 0.48 | 0.56 | 0.49 | 0.55 | 0.48 | 0.65 | 0.41 | 0.54 |
| BLEU | 0.12 | 0.22 | 0.17 | 0.20 | 0.21 | 0.10 | 0.31 | 0.12 | 0.18 |
| CrystalBLEU | 0.38 | 0.70 | 0.56 | 0.68 | 0.62 | 0.40 | 0.68 | 0.37 | 0.51 |
| AST | 0.64 | 0.60 | 0.70 | 0.57 | 0.56 | 0.66 | 0.64 | 0.50 | 0.69 |
| CodeBertScore | 0.31 | 0.52 | 0.37 | 0.32 | 0.35 | 0.49 | 0.46 | 0.31 | 0.43 |
| CodeEmbedding | 0.65 | 0.83 | 0.88 | 0.65 | 0.86 | 0.83 | 0.90 | 0.81 | 0.88 |
| CodeBLEU | 0.74 | 0.79 | 0.88 | 0.73 | 0.72 | 0.76 | 0.75 | 0.69 | 0.71 |
| **BehaveSim** | 0.64 | 0.49 | 0.79 | 0.69 | 0.44 | 0.42 | 0.46 | 0.45 | 0.64 |

Table 8: Comparison of similarity metrics on **Type 4** cases (Part 2 of 2, Cases 10-18, continued). Cases 10-14 involve sorting algorithms. Cases 15-18 involve shortest path algorithms.

| Method | 10 | 11 | 12 | 13 | 14 | 15 | 16 | 17 | 18 |
|---|---|---|---|---|---|---|---|---|---|
| ROUGE | 0.44 | 0.37 | 0.38 | 0.47 | 0.52 | 0.42 | 0.65 | 0.29 | 0.25 |
| BLEU | 0.07 | 0.08 | 0.09 | 0.14 | 0.22 | 0.14 | 0.38 | 0.07 | 0.05 |
| CrystalBLEU | 0.44 | 0.58 | 0.61 | 0.46 | 0.54 | 0.48 | 0.53 | 0.25 | 0.39 |
| AST | 0.45 | 0.39 | 0.48 | 0.65 | 0.64 | 0.50 | 0.73 | 0.49 | 0.42 |
| CodeBertScore | 0.40 | 0.39 | 0.42 | 0.25 | 0.44 | 0.28 | 0.49 | 0.30 | 0.24 |
| CodeEmbedding | 0.81 | 0.80 | 0.83 | 0.85 | 0.83 | 0.88 | 0.88 | 0.78 | 0.76 |
| CodeBLEU | 0.73 | 0.72 | 0.79 | 0.68 | 0.66 | 0.75 | 0.85 | 0.77 | 0.77 |
| **BehaveSim** | 0.44 | 0.51 | 0.36 | 0.78 | 0.29 | 0.37 | 0.35 | 0.00 | 0.05 |

## B    CHOICE OF TRAJECTORY DISTANCE MEASURES

This experiment investigates the effects of different trajectory-similarity measures on behavioral-similarity assessment. We consider four measures: mean pairwise distance, Dynamic Time Warping (DTW) distance, Edit Distance with Real Penalty (ERP), and the average cosine similarity between trajectory segments. Figure 6 illustrates the similarities among trajectories computed using these methods.

We observe that the mean pairwise distance, DTW, and ERP suggest that CG and NelderMead-Adaptive are more similar, and the cosine similarity indicates that CG and Newton-CG are more

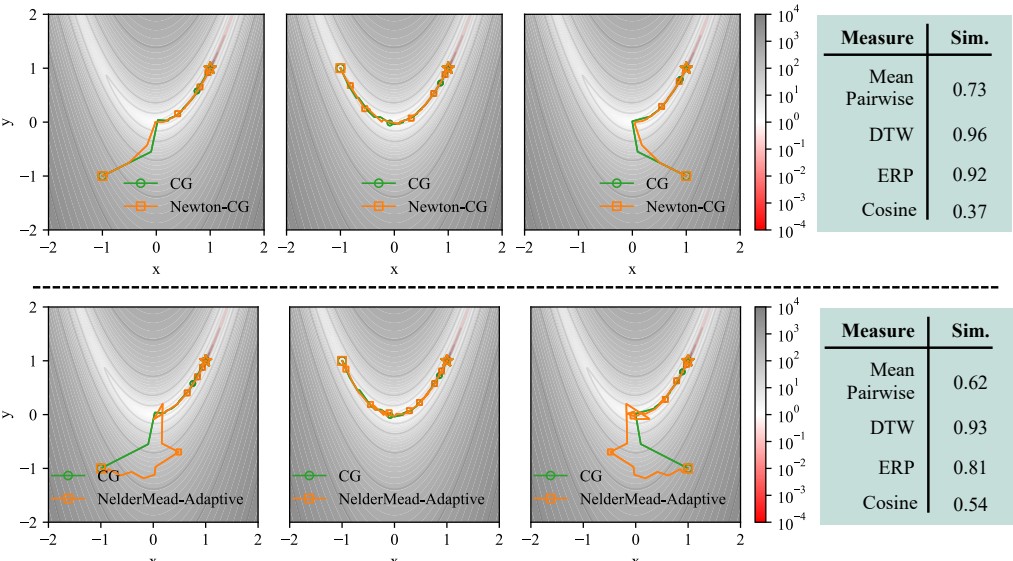

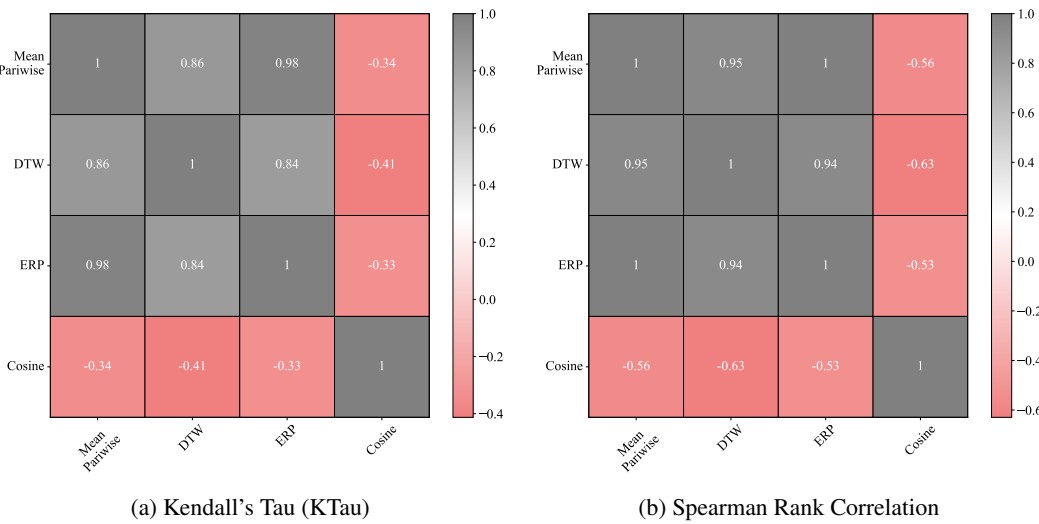

Figure 6: Demonstration of problem-solving trajectories on the Rosenbrock function.

(a) Kendall's Tau (KTau)

(b) Spearman Rank Correlation

Figure 7: Rank correlations of the similarities calculated by four trajectory similarity measures.

comparable. This implies that different trajectory similarity measures capture various aspects of the trajectories.

We also evaluate the PSTrajs of eight algorithms on the Rosenbrock function: CG, Newton-CG, SGD, Adam, BFGS, L-BFGS-B, NelderMead, and NelderMead-Adaptive. For each pair of algorithms, we compute their BehaveSim using the four trajectory similarity measures, then report Kendall's Tau (KTau) and Spearman's rank correlation coefficient to quantify the correlation. The correlation matrices are shown in Figure 7a.

The results show that mean pairwise distance, DTW, and ERP exhibit high correlations with one another, while their correlations with cosine similarity are comparatively low. This suggests that mean pairwise distance, DTW, and ERP can serve as interchangeable alternatives. In contrast, cosine similarity primarily captures the directional consistency of trajectory segments and may therefore be particularly relevant in specific application scenarios.

## C    INFLUENCES OF PSTRAJ TRUNCATION AND SAMPLING PARAMETERS ON BEHAVESIM

BehaveSim controls trajectory granularity via (i) Truncation (removing the last proportion $k$ of the trajectory) and (ii) Sampling (keeping one solution every $n$ steps). We analyze the sensitivity of similarity scores to these parameters, with results shown in Table 9.

We observe that the metric is robust to light-to-moderate processing. Truncating 10–30% of the trajectory yields scores nearly identical to the whole trajectory, and deviations only become noticeable at aggressive ratios (40–50%). Similarly, sampling intervals of 1–2 steps have minimal effect, while larger intervals (3–5) introduce expected variance. This confirms that BehaveSim is stable under reasonable parameter choices.

Table 9: Influence of trajectory sampling strategies on similarity scores.

| Settings | Type-1 | Type-2 | Type-3 | Type-4 |
|---|---|---|---|---|
| Full-Trajectory (k=0, n=0) | 0.56 | 0.73 | 1.00 | 0.46 |
| $k = 0.1, n = 0$ | 0.55 | 0.73 | 1.00 | 0.47 |
| $k = 0.2, n = 0$ | 0.55 | 0.73 | 1.00 | 0.48 |
| $k = 0.3, n = 0$ | 0.57 | 0.73 | 1.00 | 0.52 |
| $k = 0.4, n = 0$ | 0.60 | 0.73 | 1.00 | 0.58 |
| $k = 0.5, n = 0$ | 0.65 | 0.75 | 1.00 | 0.64 |
| $k = 0, n = 1$ | 0.57 | 0.72 | 1.00 | 0.52 |
| $k = 0, n = 2$ | 0.56 | 0.74 | 1.00 | 0.59 |
| $k = 0, n = 3$ | 0.56 | 0.72 | 1.00 | 0.65 |
| $k = 0, n = 4$ | 0.56 | 0.82 | 1.00 | 0.72 |
| $k = 0, n = 5$ | 0.62 | 0.82 | 1.00 | 0.74 |

## D    RELATED WORK ON LLM-AAD

The integration between large language models (LLMs) with search methods has become a prevailing paradigm in automated algorithm design (AAD) (Liu et al., 2024; Zhang et al., 2024), leading to notable advances across a spectrum of AAD applications, including mathematical discovery (Romera-Paredes et al., 2024), combinatorial optimization (Liu et al., 2024; Ye et al., 2024), Bayesian optimization (Yao et al., 2024), black-box optimization (Van Stein & Bäck, 2024), and science discovery (Shojaee et al., 2025).

Notably, recent advances have emphasized the importance of diversity in algorithm search (Romera-Paredes et al., 2024; Novikov et al., 2025). For example, methods such as FunSearch (Romera-Paredes et al., 2024) and LLM-SR (Shojaee et al., 2025) adopt a multiple-island-based program database to enhance diversity. AlphaEvolve (Novikov et al., 2025) combines the multiple-island-based model with MAP elite (Mouret & Clune, 2015) to further improve diversity. PlanSearch (Wang et al., 2024a) performs search in idea and pseudo-code spaces to increase diversity. Other methods incorporate algorithmic similarity metrics into the search process. For instance, MEoH (Yao et al., 2025) embeds AST distances into a multi-objective algorithm search, Mao et al. (2024) clusters algorithms within sub-populations according to their embedding distance. In this paper, we introduce combining BehaveSim with a search method to promote the behavioral diversity during search.

## E    IMPLEMENTATION DETAILS OF OUR SEARCH METHOD

### E.1    IMPLEMENTATION DETAILS OF FUNSEARCH+BEHAVESIM

Our search pipeline begins with a predefined "template algorithm" (detailed in Appx. § G) that provides task information for LLMs, including the algorithm's inputs and outputs, task descriptions, and argument formats. To initialize the database, $N_{\text{init}}$ randomly generated algorithms are clustered based on behavioral similarity and subsequently allocated to different islands. The search process

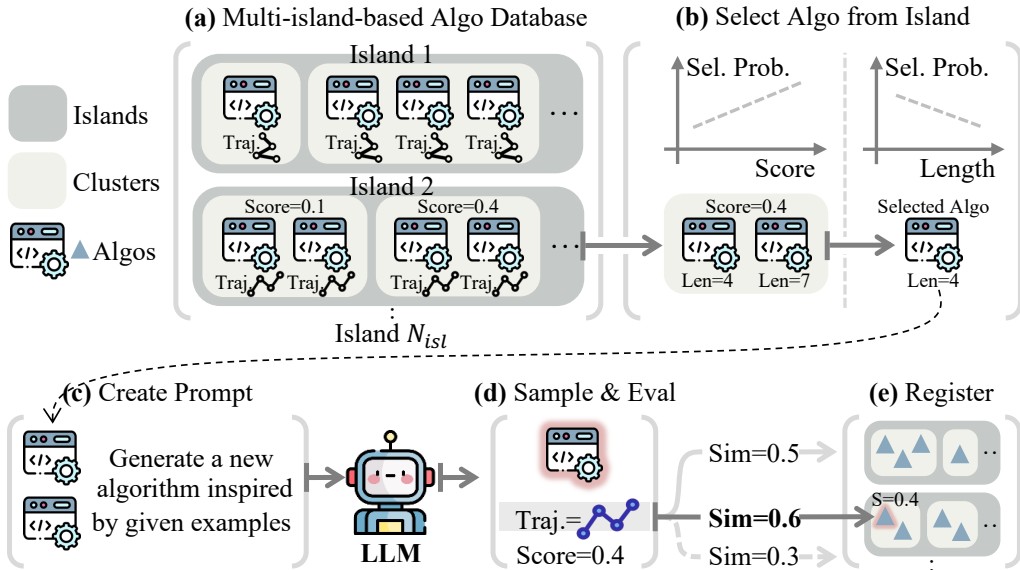

Figure 8: Overview of FunSearch+BehaveSim. **(a)** We utilize a multi-island algorithm database, where each island contains clusters of algorithms grouped by identical scores. **(b)** In the selection phase, an island is chosen from the database. Within the selected island, clusters are prioritized by score, with higher scores given preference. Once a cluster is selected, an algorithm is chosen, favoring shorter implementations to promote conciseness. **(c)** The selected algorithms are used to create a prompt, which guides the LLM to generate a new algorithm. **(d)** The newly generated algorithm is evaluated to determine its fitness score and problem-solving trajectory. **(e)** Finally, the evaluated algorithm is assigned to the island with the highest behavioral similarity and placed into the appropriate cluster.

then proceeds iteratively: algorithms are selected from a database to serve as examples for the LLM to generate new algorithms. These new algorithms are evaluated to determine their fitness score and behavioral trajectory, and are then registered back into the appropriate island in the database. The search terminates after a fixed number of $N_{\text{eval}}$ evaluations.

**Algorithm Database.** The algorithm database preserves a population of diverse algorithms obtained in the search phase. Inspired by prior works such as FunSearch (Romera-Paredes et al., 2024) and AlphaEvolve (Novikov et al., 2025), our database adopts a multiple-island-based population to promote diversity. As shown in Figure 8(a), the database consists of a fixed size of $N_{\text{isl}}$ islands, each of which groups algorithms with similar problem-solving behaviors as measured by our BehaveSim. Within each island, algorithms are further grouped into several clusters, each containing algorithms with identical fitness scores.

For database initialization, instead of cloning a single seed algorithm to each island (as in FunSearch), we first sample $N_{\text{init}} = 100$ algorithms. We then perform clustering based on their BehaveSim similarity and register these initial algorithms into the $N_{\text{isl}}$ islands.

**Algorithm Selection.** The algorithm selection phase selects two different algorithms from the algorithm database as few-shot examples. We propose two selection strategies, **S1** and **S2**, specifically:

- **Inter-island Selection (S1):** To ensure the communication of two distinct islands, **S1** strategy chooses two distinct islands in the algorithm database. We subsequently obtain an algorithm from each of them.

- **Intra-island Selection (S2):** As adopted in FunSearch, we randomly choose one island and select two distinct algorithms within it.

To balance the utility of two strategies **S1** and **S2**, we set a hyperparameter $p_{s1}$, which determines the probability of adopting the **S1** strategy in the selection phase.

As shown in Figure 8 (b), the selection of an algorithm from a given island is a two-step process. First, we select a cluster within the island, giving preference to clusters with higher fitness scores. Second, within the chosen cluster, we select an algorithm that prioritizes shorter implementations (e.g., fewer lines of code) to promote conciseness and simplicity.

**Sampling and Evaluation.** As depicted in Figure 8, once two algorithms are selected, they are formatted into a prompt for the LLM. Following a two-shot prompting approach inspired by FunSearch, we sort the examples by their fitness scores in ascending order. The prompt demonstrates that the second algorithm performs better than the first, and then instructs the LLM to generate a potentially better algorithm. We perform two queries to the LLM for each prompt to obtain two new candidate algorithms.

Algorithms sampled from the LLM are sent to a sandbox (implemented as a separate process isolated from the main process) for secure evaluation, in which we record a fitness score $s$ representing its performance of solving the specific problem, and a trajectory $t$ indicating its problem-solving behavior. We discard infeasible algorithms that either raise errors during evaluation or exceed the timeout limit.

**Register Algorithms into Algorithm Database.** After evaluation, feasible algorithms are registered in the database. A key distinction from previous methods is that we do not simply register the new algorithm back to its source island. Instead, we use BehaveSim to place it on the most similar island in the database. Given a newly evaluated algorithm with trajectory $t$ and a database of $N_{\text{isl}}$ islands, where $t_j^i$ is the trajectory of the $j$-th algorithm in island $I_i$, the target island is determined by:

$$\text{target\_island} = \underset{i \in [1, N_{\text{isl}}]}{\arg\max} \left( \frac{1}{|I_i|} \sum_{t_j^i \in I_i} \text{BehaveSim}(t, t_j^i) \right)$$

This equation calculates the average BehaveSim between the new algorithm's trajectory and all trajectories within each island and selects the island with the highest average similarity. The new algorithm is then placed in the appropriate cluster within the target island, or a new cluster is initialized if none exists for its fitness score.

**Restarting Islands.** To prevent the search from getting stuck in local optima and to manage the database size, we periodically restart a portion of the islands. Every two hours, we identify the $N_{\text{isl}}/2$ islands with the lowest-scoring best algorithms. We discard all algorithms within these islands and re-initialize them by randomly importing the best algorithms from the surviving $N_{\text{isl}}/2$ islands. This mechanism ensures a continuous influx of new ideas and prevents the database from becoming over-bloated with suboptimal solutions.

### E.2 IMPLEMENTATION DETAILS OF EOH+BEHAVESIM

For the integration of EoH+BehaveSim, we propose to augment primary performance objectives with BehaveSim to the best algorithm found (smaller is better) as another objective during the search. The integration is inspired by the success of MEoH (Yao et al., 2025), which models the AAD problem as a multi-objective optimization and employs a dominance-dissimilarity mechanism during search to enhance both diversity and search efficiency.

**Parent Selection.** As shown in Algorithm 1, the dominance matrix $D$ and dissimilarity matrix $S$ are first calculated based on the dominance relationships and BehaveSim, and $S'$ is calculated by the element-wise multiplication of $S$ and $D$. Then, the dominance-dissimilarity vector $v$ is calculated by the column-wise sum of $S'$, based on which we calculate $\pi$, which indicates the probabilities of sampling each candidate algorithm.

**Population Management.** As shown in Algorithm 2, algorithms in the old population are sorted based on their dominance-dissimilarity, and the top-$N$ algorithms survive.

**Algorithm 1:** Parent Selection

> **Input:** Population $\boldsymbol{P}$;
>   Population size $N$;
>   Parent selection size $d$
> **Output:** Selected parents $\boldsymbol{P}_{parent}$

1   $\boldsymbol{S} \leftarrow$ an $N \times N$ matrix filled with zeros
2   $\boldsymbol{D} \leftarrow$ an $N \times N$ matrix filled with zeros
3   **for** $i \leftarrow 1$ **to** $N$ **do**
4     **for** $j \leftarrow 1$ **to** $N$ **do**
5       **if** $i \neq j$ **then**
6         $\boldsymbol{S}[i,j] \leftarrow$
          $-\text{BehaveSim}(\boldsymbol{P}[i], \boldsymbol{P}[j])$
7         **if** $\boldsymbol{P}[i] \prec \boldsymbol{P}[j]$ **then**
8           $\boldsymbol{D}[i,j] \leftarrow 1$

9   $\boldsymbol{S}' \leftarrow \boldsymbol{S} \odot \boldsymbol{D}$
10   $\boldsymbol{v} \leftarrow \text{ColumnwiseSum}(\boldsymbol{S}')$
11   $\boldsymbol{\pi} \leftarrow \text{Softmax}(\boldsymbol{v})$
12   $\boldsymbol{P}_{parent} \leftarrow \text{Sample}(\boldsymbol{P}, \boldsymbol{\pi}, d)$
13   **return** $\boldsymbol{P}_{parent}$

**Algorithm 2:** Population Management

> **Input:** Population $\boldsymbol{P}$; Population size $N$
> **Output:** New population $\boldsymbol{P}'$

1   $N' \leftarrow \text{size}(\boldsymbol{P})$
2   $\boldsymbol{S} \leftarrow$ an $N' \times N'$ matrix filled with zeros
3   $\boldsymbol{D} \leftarrow$ an $N' \times N'$ matrix filled with zeros
4   **for** $i \leftarrow 1$ **to** $N$ **do**
5     **for** $j \leftarrow 1$ **to** $N$ **do**
6       **if** $i \neq j$ **then**
7         $\boldsymbol{S}[i,j] \leftarrow$
          $-\text{BehaveSim}(\boldsymbol{P}[i], \boldsymbol{P}[j])$
8         **if** $\boldsymbol{P}[i] \prec \boldsymbol{P}[j]$ **then**
9           $\boldsymbol{D}[i,j] \leftarrow 1$

10   $\boldsymbol{S}' \leftarrow \boldsymbol{S} \odot \boldsymbol{D}$
11   $\boldsymbol{v} \leftarrow \text{ColumnwiseSum}(\boldsymbol{S}')$
12   $\boldsymbol{k} \leftarrow \text{DescendingSortedIndexes}(\boldsymbol{v})$
13   $\boldsymbol{P}' \leftarrow \emptyset$
14   **for** $i \leftarrow 1$ **to** $N$ **do**
15     $\boldsymbol{P}' \leftarrow \boldsymbol{P}' \cup \boldsymbol{P}[\boldsymbol{k}[i]]$
16   **return** $\boldsymbol{P}'$

# F   EXPERIMENT DETAILS AND ANALYSIS

## F.1   SETTINGS FOR AAD TASKS AND METHODS

**Admissible Set Problem (ASP).**   ASP aims to maximize the size of the set while fulfilling the criteria below: (1) The elements of the set are vectors belonging to $\{0, 1, 2\}^n$. (2) Each vector has the same number $w$ of non-zero elements but a unique support. (3) For any three distinct vectors, there is a coordinate in which their three respective values are $\{0, 1, 2\}, \{0, 1, 2\}, \{0, 1, 2\}$. Following prior works (Romera-Paredes et al., 2024), we set $n = 15$ and $w = 10$ in this work.

**Partial Solutions in ASP.**   The objective is to design a priority function that scores candidate vectors. At each step, the highest-scoring vector is added to the set, forming a partial solution. The distance between two partial solutions is defined as the edit distance between their current sets.

**Traveling Salesman Problem (TSP).**   TSP aims to find a route that minimizes the total distance traveled by a salesman who must visit each city exactly once before returning to the starting point. We investigate the constructive heuristic design for TSP (Matai et al., 2010). Specifically, we adopt an iteratively constructive framework that starts from a single node, iteratively selects the next node until all nodes have been selected, and then returns to the start node. The task is to design a heuristic for choosing the next node to minimize the route length. We generate five TSP instances, each comprising 50 cities, for training.

**Partial Solutions in TSP.**   At each step, the algorithm selects the next city, yielding a progressively constructed route. A partial solution is an ordered list of visited cities. The distance between two solutions is the edit distance between their routes, with similarity aggregated across five instances.

**Settings for LLM-AAD Methods.**   For FunSearch, we use 10 islands for the database. Each prompt presents two reference algorithms, and we sample two algorithms per prompt. For EoH, the population size is set to 20. To ensure fairness, EoH is terminated based on the total evaluation budget instead of the maximum number of generations. All methods are evaluated under identical evaluation budgets and stopping criteria for a fair comparison.

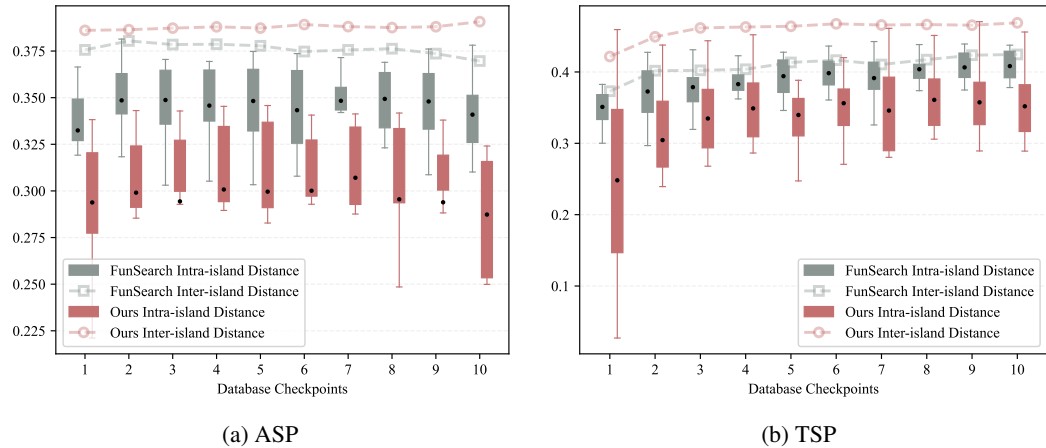

(a) ASP                                        (b) TSP

Figure 9: Comparison of intra- and inter-island distances for FunSearch and our method. Each box shows the distribution of intra-island distances across 10 islands at each checkpoint, while the curves track the average inter-island distance. Lower distances indicate similar problem-solving behaviors; higher distances indicate more diverse behaviors.

## F.2 ANALYSIS OF DIVERSITY

**Motivations.** A core challenge in the search process is to balance exploration across diverse solution spaces with exploitation within promising regions. We hypothesize that our method, with an optimized algorithm database, can achieve a more effective balance than FunSearch. To verify this, we analyze the behavioral diversity of algorithms within and across islands. We compare our method against FunSearch under an identical multi-island database setup, where the only difference is the population management strategy. At each database checkpoint, we sample $N = 50$ algorithms from each of the 10 islands to serve as prototypes.

**Intra-island distance.** This metric quantifies the behavioral coherence within an island, reflecting local exploitation. For a set of representative trajectories $P_k = \{t_1, \ldots, t_N\}$ from an island $k$, the distance is:

$$D_{\text{intra}}(P_k) = \frac{1}{\binom{N}{2}} \sum_{1 \leq i < j \leq N} \left(1 - \text{BehaveSim}(t_i, t_j)\right), \quad t_i, t_j \in P_k \tag{1}$$

**Inter-island distance.** This metric measures the behavioral separation between islands, indicating global exploration. For two distinct islands $k$ and $l$, the distance is:

$$D_{\text{inter}}(P_k, P_l) = \frac{1}{N^2} \sum_{t_i \in P_k} \sum_{t_j \in P_l} \left(1 - \text{BehaveSim}(t_i, t_j)\right) \tag{2}$$

Figure 9 visualizes these metrics, where boxes show the distribution of intra-island distances over all islands, and the curves track the average inter-island distance. We observe that:

- Our method yields consistently lower intra-island distances (red boxes vs. green). This indicates that each island maintains a more behaviorally coherent population, enabling focused exploitation.

- Simultaneously, our method achieves higher inter-island distances (red curves vs. green). This demonstrates that the islands are more behaviorally distinct, promoting global exploration across diverse problem-solving strategies.

These results reveal that our method achieves a superior balance between exploitation and exploration, a crucial factor for robust discovery of novel, high-performing algorithms.

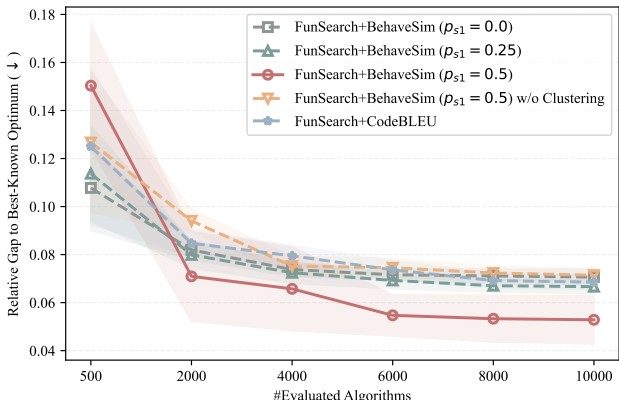

Figure 10: Ablation study on ASP. We evaluate the effect of inter-island selection probability $p_{s1}$ and the clustering operation in the initialization of the algorithm database.

### F.3 ABLATION STUDY

To better understand the contribution of each component in our framework, we conduct ablation studies on the ASP task. We focus on two factors: (i) the probability $p_{s1}$ of adopting the inter-island selection strategy (**S1**) versus the intra-island strategy (**S2**), and (ii) the effect of clustering within islands during algorithm database construction. The results are shown in Figure 10.

**Effect of Inter-Island Selection ($p_{s1}$).** We vary $p_{s1}$ from 0 to 0.5. When $p_{s1} = 0$ (purely intra-island selection), the method degenerates into a variant similar to FunSearch, which restricts algorithm mixing to within a single island. Increasing $p_{s1}$ consistently improves performance, with the best results obtained at $p_{s1} = 0.5$. This demonstrates the importance of promoting cross-island communication: algorithms from different islands capture diverse problem-solving behaviors, and combining them enhances exploration of the search space.

**Effect of Clustering.** We also evaluate a variant without clustering during database initialization. As shown in Figure 10, the absence of clustering significantly degrades performance, even when inter-island selection is enabled ($p_{s1} = 0.5$).

## G TEMPLATE ALGORITHM

We initialize all methods except for EoH with identical template algorithms to ensure fairness. The template algorithms employed in the ASP and TSP tasks are shown in the following listings.

Code 1: Template Algorithm for ASP

```python
import math
import numpy as np

def priority(el: tuple[int, ...], n: int, w: int) -> float:
    """
    Returns the priority with which we want to add 'el' to the set.
    Args:
        el: the unique vector has the same number w of non-zero elements.
        n : length of the vector.
        w : number of non-zero elements.
    """
    return 0.
```

Code 2: Template Algorithm for TSP

```python
import numpy as np
```

```python
def select_next_node(
    current_node: int,
    destination_node: int,
    unvisited_nodes: np.ndarray,
    distance_matrix: np.ndarray
) -> int:
    """
    Design a novel algorithm to select the next node in each step.
    Args:
        current_node: ID of the current node.
        destination_node: ID of the destination node.
        unvisited_nodes: Array of IDs of unvisited nodes.
        distance_matrix: Distance matrix of nodes.

    Return:
        ID of the next node to visit.
    """
    next_node = unvisited_nodes[0]
    return next_node
```

Code 3: Template Algorithm for CPP

```python
import numpy as np
import math

def pack_circles(n: int) -> np.ndarray:
    """
    Pack n circles in a unit square to maximize sum of radii.

    Args:
        n: Number of circles to pack
    Returns:
        Numpy array of shape (n, 3) where each row is (x, y, radius)
        All values should be between 0 and 1
        Circles must not overlap

    Important: Set "all" random seeds to 2025, including the packages (
        such as scipy sub-packages) involving random seeds.
    """
    np.random.seed(2025)
    if n == 0:
        return np.zeros((0, 3))

    circles = np.zeros((n, 3))

    # Place first circle in the center
    circles[0] = [0.5, 0.5, 0.5]

    for i in range(1, n):
        max_r = 0
        best_pos = (0.5, 0.5)

        # Generate candidate positions near existing circles and
            boundaries
        candidates = []
        for j in range(i):
            cx, cy, r = circles[j]
            for angle in np.linspace(0, 2*np.pi, 20):
                dx = np.cos(angle)
                dy = np.sin(angle)
                new_x = cx + (r + 0.01) * dx
                new_y = cy + (r + 0.01) * dy
                if 0 <= new_x <= 1 and 0 <= new_y <= 1:
                    candidates.append((new_x, new_y))
```

```
41
42          # Add boundary points
43          for b in np.linspace(0, 1, 20):
44              candidates.extend([(b, 0), (b, 1), (0, b), (1, b)])
45
46          # Evaluate each candidate
47          for (cx, cy) in candidates:
48              # Calculate maximum possible radius
49              current_r = min(cx, 1 - cx, cy, 1 - cy)
50
51              # Check against existing circles
52              if i > 0:
53                  existing = circles[:i, :2]
54                  radii = circles[:i, 2]
55                  distances = np.sqrt((existing[:, 0] - cx)**2 + (existing
                        [:, 1] - cy)**2)
56                  min_dist = np.min(distances - radii)
57                  current_r = min(current_r, min_dist)
58
59              if current_r > max_r:
60                  max_r = current_r
61                  best_pos = (cx, cy)
62
63          circles[i] = [best_pos[0], best_pos[1], max_r]
64
65      return circles
```

## H    USE OF LARGE LANGUAGE MODELS

Large Language Models are employed in two ways in this work. (i) We use LLMs to aid or polish writing in this paper. (ii) This work investigates LLM-based automated algorithm design. Therefore, LLMs are used in the experiments.

