# OpenReview forum: "Rethinking Code Similarity for Automated Algorithm Design with LLMs"
_ICLR.cc/2026/Conference — ICLR 2026 Poster_

### Official Review · Reviewer_TTPL · 2025-10-29

**Soundness:** 3
**Presentation:** 4
**Contribution:** 3
**Rating:** 6
**Confidence:** 3

**Summary:**

This paper notes existing code similarity metrics (e.g., CodeBLEU) fail to reflect algorithm similarity in LLM-AAD. It proposes BehaveSim, which measures similarity via algorithms’ problem-solving trajectories. Using DTW to align trajectories and normalize distances, BehaveSim outperforms traditional metrics in validation. It enhances LLM-AAD performance and aids algorithm analysis, with limitations on non-iterative algorithms.

**Strengths:**

The perspective of the paper is quite novel, as it calculates similarity based on the problem-solving trajectories of algorithms. The paper argues that traditional code similarity metrics fail to reflect true algorithm similarity, so it proposes BehaveSim, which quantifies behavioral similarity by analyzing sequences of intermediate solutions (trajectories) generated during algorithm execution.

**Weaknesses:**

The proposed behavioral similarity metric, is only designed for iterative algorithms (e.g., sorting, optimization algorithms) and cannot be directly applied to other types of algorithms like machine learning models.

**Questions:**

1. Regarding the differences in trajectory lengths among different iterative algorithms, what methods are adopted in the paper to avoid biases in BehaveSim's similarity calculation?

2. How is the hyperparameter p_s1 (probability of inter-island selection) tuned in the paper, and what empirical evidence supports that p_s1=0.5 is the optimal value for balancing exploration and exploitation?

---

> ### Author Response · Authors · 2025-11-21
> **Response to Reviewer TTPL**
>
> > **Weakness 1:** The proposed behavioral similarity metric, is only designed for iterative algorithms (e.g., sorting, optimization algorithms) and cannot be directly applied to other types of algorithms like machine learning models.
>
> We acknowledge that the imprecise wording in our scope of "iterative algorithm" may have caused confusion for the reviewer. We have revised the description to more accurately reflect the intended meaning.
>
> ### Scope of iterative algorithms
>
> In this work, the scope of "iterative algorithm" encompasses methods that progressively refine intermediate solutions through repeated computation, rather than solving analytically in a single step. Formally, we express this as $x_{t+1} = f(x_t)$, where $x_t$ represents the solution (or partial solution) at step $t$, and $f$ is the iterative algorithm mapping $x_t$ to $x_{t+1}$.
>
> ### BehaveSim for ML algorithms
>
> Many ML algorithms fall naturally within this scope. For instance, in *K-Means*, $x_t$ represents cluster assignments and centroids, with $f$ performing the update. Similarly, in Reinforcement Learning (RL), $x_t$ corresponds to policy parameters or agent states evolving via environment interactions. Both examples generate solution trajectories fully compatible with BehaveSim.
>
>
>
>
>
> > **Question 1:** Regarding the differences in trajectory lengths among different iterative algorithms, what methods are adopted in the paper to avoid biases in BehaveSim's similarity calculation?
>
> We thank the reviewer for raising this question. As shown in lines 300-304 in our manuscript, DTW has the primary advantage of allowing flexible temporal matching to align trajectories of varying lengths. This ensures that similarity is computed based on behavioral patterns rather than on strict step-by-step alignment.
>
> Formally, given two trajectories $X = (x_1, \dots, x_m)$ and $Y = (y_1, \dots, y_n)$, the DTW distance is defined as:
>
> $$
> D(i, j) = d(x_i, y_j) + \min \begin{cases}
> D(i-1, j-1) \\\\
> D(i-1, j)   \\\\
> D(i, j-1)
> \end{cases}
> $$
>
> where $\mathcal{A}(m, n)$ denotes the set of all possible alignment paths between the two trajectories.
>
> This minimization is achieved using a dynamic programming approach. A cost matrix $D$ is constructed, where $D(i, j)$ represents the minimum cumulative distance to align the sub-trajectories $(x_1, \dots, x_i)$ and $(y_1, \dots, y_j)$. The recursive formula is:
>
> $$
> D(i, j) = d(x_i, y_j) + \min \begin{cases} D(i-1, j-1) \\ D(i-1, j)  \\ D(i, j-1)  \end{cases}
> $$
>
> This recursive formula is the key to DTW's ability to handle temporal shifts. The $\min$ function allows the alignment path $\pi$ to adjust vertically ($D(i-1, j)$) and horizontally ($D(i, j-1)$) in the cost matrix.
>
> This enables a single solution in one trajectory (e.g., $x_i$) to be aligned with multiple consecutive solutions in the other trajectory (e.g., $y_{j-k}, \dots, y_j$), or vice versa.
>
>
>
>
>
> > **Question 2:** How is the hyperparameter p_s1 (probability of inter-island selection) tuned in the paper, and what empirical evidence supports that p_s1=0.5 is the optimal value for balancing exploration and exploitation?
>
> We thank the reviewer for raising this insightful question. In our FunSearch+BehaveSim method, the hyperparameter $p_{s1}$ controls the relative frequency of the S1 (inter-island) and S2 (intra-island) selection strategies, which influences the degree of cross-island information exchange versus within-island refinement.
>
> To select $p_{s1}$, we conducted an empirical ablation study (Appendix Figure 10 in the manuscript), evaluating several values ($0.0$, $0.25$, $0.5$). The results show that $p_{s1}=0.5$ achieves the best search performance. While our goal is not to claim that $p_{s1}=0.5$ is theoretically optimal for achieving the optimal balance between exploration and exploitation, the ablation study suggests that this value of $p_{s1}=0.5$ enables sufficient cross-island communication for achieving promising search performance.

---

> ### Comment · Reviewer_TTPL · 2025-11-26
>
> I appreciate the authors' work during the rebuttal period to address my concerns. I have chosen to keep my original positive rating.

---

### Official Review · Reviewer_KsNQ · 2025-10-31

**Soundness:** 3
**Presentation:** 2
**Contribution:** 3
**Rating:** 6
**Confidence:** 3

**Summary:**

This paper introduces BehaveSim, a novel similarity metric designed to measure algorithm similarity from a behavioral perspective, rather than code-level or output-level similarity. The authors argue that existing metrics (token-, AST-, embedding-, or execution-based) fail to capture the problem-solving behavior of algorithms—especially in the context of LLM-based Automated Algorithm Design (LLM-AAD), where generated code can differ syntactically yet implement equivalent ideas.
BehaveSim represents each algorithm as a problem-solving trajectory, i.e., a sequence of intermediate solutions generated during execution. The similarity between two algorithms is then defined as the resemblance between their trajectories, computed via Dynamic Time Warping (DTW). Experiments show that BehaveSim better differentiates between algorithms that have similar code but distinct behavior (e.g., BFS vs DFS, insertion sort vs bubble sort), and vice versa.
The paper further demonstrates two applications: (1) improving behavioral diversity in LLM-AAD search (enhancing FunSearch performance on ASP and TSP tasks), and (2) clustering algorithm behaviors for interpretability and discovery.

**Strengths:**

Novel Perspective:
The paper identifies a clear conceptual gap between code similarity and algorithmic behavior similarity, proposing an elegant behavioral abstraction based on execution trajectories. This reframing is insightful and well-motivated in the context of LLM-generated algorithms.

Concrete Implementation (BehaveSim):
The definition of behavioral trajectories and the use of DTW distance provide a simple yet powerful operationalization of behavioral similarity. The methodology is well formalized, reproducible, and extensible.

Comprehensive Benchmark:
The authors curate a systematic dataset with four categories (Type-1 to Type-4) decoupling code-, behavior-, and result-level similarities, offering a rigorous evaluation against existing metrics such as CodeBLEU, CodeBERTScore, and execution-based scores.

Strong Empirical Results:
BehaveSim achieves intuitive and consistent performance across all dataset types (e.g., correctly scoring 1.0 for Type-3 pairs with equivalent behaviors). Integration with FunSearch also improves both convergence and final performance on ASP and TSP benchmarks, validating practical relevance.

Interpretability and Analysis:
The algorithm clustering experiment (Fig. 5) compellingly illustrates how BehaveSim distinguishes semantically similar but syntactically different implementations, supporting new interpretability avenues in algorithm discovery.

**Weaknesses:**

Scope Limitation:
BehaveSim applies only to iterative algorithms producing discrete trajectories. Many LLM-generated algorithms, including stochastic, differentiable, or recursive paradigms, are excluded. This significantly restricts generality.

Metric Design Choices:
The use of DTW on normalized edit or Euclidean distances is heuristic; there’s limited justification for why DTW best captures “behavioral similarity.” Ablation on alternative measures (ERP, cosine, etc.) is included but not theoretically grounded.

Dependence on Trajectory Definition:
Defining what constitutes a “partial solution” may require manual instrumentation of each algorithm, limiting scalability and automation for arbitrary code. This makes BehaveSim less plug-and-play for general LLM evaluation pipelines.

Comparative Baseline Gaps:
Although the work compares to standard code metrics, it lacks comparison to semantic or dynamic program analysis methods (e.g., symbolic execution traces, graph-based semantic embeddings). These could offer a fairer baseline.

Moderate Empirical Gains:
While improvements in AAD tasks (Table 3) are consistent, they are modest (~10–15% relative gap reduction) and limited to small benchmark scales. Broader tests on complex algorithmic synthesis domains would strengthen impact.

Overclaiming “Novelty” for AAD:
Integrating diversity measures into LLM-AAD (FunSearch + BehaveSim) is conceptually incremental to existing multi-island or MAP-Elite-based diversity frameworks. The true novelty lies more in behavioral similarity than in AAD improvement.

**Questions:**

N/A

---

> ### Author Response · Authors · 2025-11-21
> **Response to Reviewer KsNQ**
>
> > **Weakness 1:** Scope Limitation: BehaveSim applies only to iterative algorithms producing discrete trajectories. Many LLM-generated algorithms, including stochastic, differentiable, or recursive paradigms, are excluded. This significantly restricts generality.
>
> We acknowledge that the imprecise wording in our scope of "iterative algorithm" may have caused confusion for the reviewer. We have revised the description to more accurately reflect the intended meaning.
>
> ### Scope of iterative algorithms
>
> In this work, we use the term "iterative algorithm" to denote a class of algorithms whose solutions cannot be obtained analytically in a single step. Instead, they produce a sequence of intermediate solutions and progressively refine the solution through repeated computation. Formally, we express such algorithms as $x_{t+1} = f(x_t)$, where $x_t$ represents the solution (or partial solution) at step $t$, and $f$ is the iterative algorithm that maps the current solution $x_t$ to the next solution $x_{t+1}$.
>
> Under this scope, the algorithm families mentioned by the reviewer, i.e, stochastic algorithms, differentiable algorithms, and recursive algorithms, naturally fall within the scope of BehaveSim.
>
> ### BehaveSim for stochastic algorithms
>
> Stochastic intermediate states do not invalidate the concept of behavioral similarity, but we require a new way to measure the pairwise distance.
>
> 1. **Sample multiple solutions at each step.** If the transition $x_{t+1}=f_A(x_t \mid \xi)$ involves a random noise $\xi$, we obtain multiple samples to approximate the distribution $P_{t+1}(x)$ and similarly obtain $P_{t+1}(y)$ for algorithm $B$. The pairwise distance can then be defined as a divergence $D(P_{t+1}(x), P_{t+1}(y))$
>
> 2. **Fixing random seed.** Fixing the random seed is another way to reduce the randomness and ensure reproducibility.
>
> Both methods will further aggregate similarity across multiple problem instances (as mentioned in Sec. 3, lines 353-360 in our manuscript), which naturally reduces variance induced by randomness.
>
> Sampling multiple solutions at each step is more robust since it compares distributions, but it requires more computational overhead for sampling. Fixing random seeds ensures reproducibility, but may lead to biased calculations of behavioral similarity. Depending on the specific context, either approach can be selected accordingly.
>
> ### Using BehaveSim for differentiable algorithms
>
> BehaveSim extends naturally to differentiable algorithms, such as model training via SGD. In this context, the solution $x_t$ corresponds to the model weights $\theta_t$, and the algorithm $f$ represents the gradient-based update step producing $x_{t+1}$. Consequently, the sequence of evolving weights constitutes a valid problem-solving trajectory for BehaveSim analysis.
>
> ### Using BehaveSim for recursive algorithms
>
> Recursive procedures fit this scope by treating recursive calls as state transitions. For instance, the recursive DFS in Figure 1(a) updates the traversal route (partial solution) at each step, forming a problem-solving trajectory trackable by BehaveSim.
>
>
>
>
>
> > **Weakness 2:** Metric Design Choices: The use of DTW on normalized edit or Euclidean distances is heuristic; there's limited justification for why DTW best captures "behavioral similarity". Ablation on alternative measures (ERP, cosine, etc.) is included but not theoretically grounded.
>
> We thank the reviewer for highlighting this point. We clarify that our primary objective is not to claim theoretical optimality for DTW, but to provide a tangible method for measuring problem-solving behavior via trajectories.
>
> In this context, DTW is a natural choice as it handles temporal misalignment and variable-length trajectories, which arise when algorithms converge at different rates or follow paths of varying lengths. This makes DTW particularly suitable for comparing such trajectories. Moreover, empirical results in Figures 3 and 6 in our manuscript demonstrate that DTW captures behavioral similarity in a way that aligns with human intuition.
>
> While we acknowledge that the choice of distance metric is important, our focus in this work is on introducing a tangible way for measuring problem-solving behavior, rather than establishing a theoretically optimal similarity measure.

---

> ### Author Response · Authors · 2025-11-21
> **Response to Reviewer KsNQ**
>
> > **Weakness 3:** Dependence on Trajectory Definition: Defining what constitutes a "partial solution" may require manual instrumentation of each algorithm, limiting scalability and automation for arbitrary code. This makes BehaveSim less plug-and-play for general LLM evaluation pipelines.
>
> We thank the reviewer for this important question and clarify the following points regarding instrumentation and automation.
>
> ### On whether defining "partial solutions" requires manual instrumentation
>
> We clarify that BehaveSim requires no algorithm-specific instrumentation. The definition of a solution (or partial solution) is derived from the problem formulation itself, ensuring a uniform interface $x_{t+1} = f(x_t)$ across all algorithms addressing the same task. For example, in TSP, the structure of a partial tour is standard regardless of the specific algorithm. Thus, BehaveSim records solution transitions without requiring per-algorithm manual adjustments.
>
> ### On scalability and automation for arbitrary code
>
> Theoretically, BehaveSim can measure behavioral similarity for any algorithm that can be expressed in the form $x_{t+1} = f(x_t)$, allowing us to capture its problem-solving trajectory.
>
> In practice, however, applying BehaveSim to arbitrary code remains challenging. For algorithms implemented through the "ask-and-tell" interface (ask the algorithm for new candidate(s), and tell the algorithm the fitness to update state), the intermediate states are directly observable, making it straightforward to compute BehaveSim. For other implementations, as long as the intermediate results can be explicitly obtained, BehaveSim can also be applied.
>
> Extending the applicability of BehaveSim to a broader class of algorithms is an important direction for future work, which will enhance its practical utility in more use cases.

---

> ### Author Response · Authors · 2025-11-21
> **Response to Reviewer KsNQ**
>
> > **Weakness 4:** Comparative Baseline Gaps: Although the work compares to standard code metrics, it lacks comparison to semantic or dynamic program analysis methods (e.g., symbolic execution traces, graph-based semantic embeddings). These could offer a fairer baseline.
>
> We thank the reviewer for the suggestion. We conduct additional analyses below.
>
> ### Comparison to execution-trace-based methods
>
> From a methodological perspective, existing execution-trace-based approaches [1-3] track low-level internal states, such as variable values, instruction-level changes, or memory attributes.
>
> In contrast, the problem-solving trajectory we use can be viewed as a highly specialized trace, corresponding to tracking the variation of a solution or partial solution in the algorithm.
>
> For general-purpose program analysis, traces for multiple variables and function calls indeed contain rich and useful information. However, for characterizing problem-solving behavior, the large amount of low-level state changes (multiple variable states, function-call logs, etc.) may introduce excessive information unrelated to the algorithm's problem-solving dynamics, which could potentially obscure similarity calculations.
>
> Empirically, we re-implemented NeXT [1] using `pysnooper` [4] to trace internal variable dynamics. Since NeXT does not define a similarity metric, we converted the traces into tokens and embeddings, measuring similarity via N-Gram-based methods (e.g., BLEU) and embedding models (Jina-Code-Embedding, Qwen3-Embedding-0.6B).
>
> As shown in Table 1 below, these methods exhibit high similarity scores for dissimilar behaviors (Types 1 and 2), indicating that they fail to distinguish behavioral differences. This confirms that measuring similarity via raw execution traces is less effective than BehaveSim's problem-solving trajectory, which comprises intermediate solutions.
>
> ### Comparison to semantic-based methods
>
> Our manuscript already included comparisons with semantic embeddings, such as *CodeBERTScore* and *Jina-Code-Embedding* (this model is referred to as "CodeEmbedding" in our manuscript). We additionally evaluated a semantic embedding model, *Qwen3-Embedding-0.6B*. As shown in Table 1, we consistently observed that these models fail to capture behavioral similarity.
>
> These additional results reinforce our main claim: neither semantic embeddings nor low-level execution traces reliably reflect algorithm behavior. BehaveSim's explicit focus on the dynamics of problem-solving solutions makes it a more suitable method for this purpose.
>
>
>
> **Table 1:** Average similarity on four types of data calculated by various metrics. Methods labeled with "*" are newly added experiments.
>
> | **Method Type**                     | **Method Name**          | **Type-1** | **Type-2** | **Type-3** | **Type-4** |
> | ----------------------------------- | ------------------------ | ---------- | ---------- | ---------- | ---------- |
> | Based on Token Match                | *ROUGE*                  | 0.95       | 0.96       | 0.70       | 0.47       |
> |                                     | *BLEU*                   | 0.83       | 0.94       | 0.42       | 0.16       |
> |                                     | *CrystalBLEU*            | 0.97       | 0.99       | 0.68       | 0.51       |
> | Based on Structure                  | *AST*                    | 0.96       | 1.00       | 0.76       | 0.57       |
> | Combine Token Match and Structure   | *CodeBLEU*               | 0.97       | 0.94       | 0.91       | 0.75       |
> | Based on Embedding                  | *CodeBertScore*          | 0.84       | 0.97       | 0.60       | 0.38       |
> |                                     | *Jina-Code-Embedding*    | 0.99       | 0.99       | 0.90       | 0.84       |
> |                                     | *Qwen3-Embedding-0.6B*\* | 0.94       | 0.93       | 0.87       | 0.73       |
> | Based on Execution Results          | --                       | 1.00       | 0.00       | 1.00       | 1.00       |
> | Based on Execution Trace            | *BLEU Similarity*\*      | 0.86       | 0.95       | 0.61       | 0.54       |
> |                                     | *Jina-Code-Embedding*\*  | 1.00       | 1.00       | 0.87       | 0.77       |
> |                                     | *Qwen3-Embedding-0.6B*\* | 0.99       | 1.00       | 0.91       | 0.78       |
> | Similarity of their behavior (Ours) | *BehaveSim*              | 0.56       | 0.73       | 1.00       | 0.46       |
>
>
>
> **References:**
>
> 1. Ni et. al. NExT: Teaching Large Language Models to Reason about Code Execution. ICML 2024.
> 2. Pei et. al. TREX: Learning Execution Semantics from Micro-Traces for Binary Similarity. IEEE Transactions on Software Engineering. 2023.
> 3. Wang et. al. Combining Structured Static Code Information and Dynamic Symbolic Traces for Software Vulnerability Prediction. ICSE 2024.
> 4. Rachum et. al. PySnooper: Never use print for debugging again. URL https://github.com/cool-RR/PySnooper. 2019.

---

> ### Author Response · Authors · 2025-11-21
> **Response to Reviewer KsNQ**
>
> > **Weakness 5:** Moderate Empirical Gains: While improvements in AAD tasks (Table 3) are consistent, they are modest ($\sim 10–15\%$ relative gap reduction) and limited to small benchmark scales. Broader tests on complex algorithmic synthesis domains would strengthen impact.
>
> We thank the reviewer for the constructive feedback.
>
> ### Clarification on the modest performance gains
>
> Our evaluation focuses on the relative improvement provided by BehaveSim. As shown in Table 2, *FunSearch+BehaveSim* achieves significant gains over the original *FunSearch*:
>
> - **ASP:** 10.3% (top-1) and 14.1% (top-10) relative gap reduction;
> - **TSP:** 31.1% (top-1) and 34.2% (top-10) relative gap reduction.
>
> We also additionally incorporate BehaveSim within *EoH*. As shown in Table 2, *EoH+BehaveSim* also outperforms its original counterpart:
>
> - **ASP:** 11.0% (top-1) and 8.1% (top-10) relative gap reduction;
> - **TSP:** 12.2% (top-1) and 12.6% (top-10) relative gap reduction.
>
> These results confirm that encouraging behavioral diversity in LLM-AAD leads to consistent and meaningful performance improvements.
>
> **Table 2:** Performance of the top-1 and top-10 algorithms obtained by each AAD method. Metrics are the relative gap to the best-known optimum (%), reported as the mean $\pm$ standard deviation over three runs (lower is better). Best results are in **bold**; second-best are $\underline{underlined}$.
>
> | |**ASP** (Top-1)| **ASP** (Top-10)|**TSP** (Top-1)|**TSP** (Top-10)|
> | - | - | - | - | - |
> |*FunSearch*| $5.84_{\pm 0.55}$| $6.15_{\pm 0.59}$ |$25.22_{\pm 0.56}$|$27.46_{\pm 1.21}$|
> | *FunSearch+BehaveSim*| $\textbf{5.24}_{\pm 1.05}$| $\textbf{5.28}_{\pm 1.01}$| $\underline{17.37_{\pm 0.23}}$ | $\underline{18.05_{\pm 0.42}}$|
> | *EoH*| $5.99_{\pm 0.20}$| $6.08_{\pm 0.31}$ | $18.86_{\pm 0.20}$| $19.68_{\pm 0.42}$ |
> | *EoH+BehaveSim*| $\underline{5.33_{\pm 0.85}}$ | $\underline{5.59_{\pm 0.91}}$ | $\textbf{16.55}_{\pm 2.80}$| $\textbf{17.20}_{\pm 3.32}$|
>
> ### Clarification on the benchmark scales
> Regarding TSP, we acknowledge the relatively small scale of the original experiments and have conducted evaluations on larger instances. In the original manuscript, the TSP experiments were performed on problem instances with 50 cities. We have now further tested the "pa561" and "pa1002" instances from TSPLib, which contain 561 and 1,002 cities, respectively. The corresponding results are provided in Table 3. These additional experiments demonstrate that FunSearch+BehaveSim remains effective on substantially larger TSP problems.
>
> **Table 3:** Performance comparison on Top-1 and Top-10 performance on two TSP instances. Metrics are the relative gap to the best-known optimum (%), reported as the mean $\pm$ standard deviation over three runs (lower is better). Best results are in **bold**.
>
> | | **pa561 (Top-1)**   | **pa561 (Top-10)** | **pr1002 (Top-1)**  | **pr1002 (Top-10)**   |
> | - | -| - | - | - |
> | *FunSearch* | $37.01_{\pm 2.76}$  | $43.41_{\pm 6.00}$  | $19.50_{\pm 0.88}$  | $21.34_{\pm 0.94}$  |
> | *FunSearch+BehaveSim* | $\textbf{27.11}_{\pm 1.48}$ | **$\textbf{42.32}_{\pm 6.92}$** | **$\textbf{18.69}_{\pm 0.49}$** | **$\textbf{20.86}_{\pm 1.54}$** |
>
>
> However, we emphasize the substantial complexity of our ASP benchmark: FunSearch evaluates 2.5 million candidate algorithms to solve this task [5].
>
> To further test the capability on complex tasks, we included the Circle Packing Problem (CPP) [6].
>
> As shown in Table 4, both FunSearch+BehaveSim and EoH+BehaveSim consistently outperform their original counterparts on CPP. This reinforces the effectiveness of our method in solving challenging AAD problems.
>
>
> **Table 4:** Performance comparison on Top-1 and Top-10 performance on the Circle Packing Problem. The performance is determined by the sum of radii aggregated over three independent runs, with the higher the better. Results are reported as Mean $\pm$ Std. Best results are in **bold**; second-best are $\underline{underlined}$.
>
> |  | **CPP** (Top-1)  | **CPP** (Top-10)|
> | - | - | - |
> | FunSearch | $2.6259_{\pm 0.0051}$ | $2.6127_{\pm 0.0143}$ |
> | FunSearch + BehaveSim | $\mathbf{2.6293}_{\pm 0.0026}$ | $\mathbf{2.6249}_{\pm 0.0036}$ |
> | EoH | $2.5724_{\pm 0.0482}$  | $2.5535_{\pm 0.0562}$ |
> | *EoH+BehaveSim* | $\underline{2.6263_{\pm 0.0054}}$ | $\underline{2.6234_{\pm 0.0066}}$ |
>
>
>
> **References:**
>
> 5. Romera-Paredes et. al. Mathematical Discoveries from Program Search with Large Language Models. Nature. 2024.
> 6. Alexander Novikov et. al. AlphaEvolve: A Coding Agent for Scientific and Algorithmic Discovery. URL https://arxiv.org/abs/2506.13131.

---

> ### Author Response · Authors · 2025-11-21
> **Response to Reviewer KsNQ**
>
> > **Weakness 6:** Overclaiming "Novelty" for AAD: Integrating diversity measures into LLM-AAD (FunSearch+BehaveSim) is conceptually incremental to existing multi-island or MAP-Elite-based diversity frameworks. The true novelty lies more in behavioral similarity than in AAD improvement.
>
> We thank the reviewer for the comments. Our intention is not to claim conceptual novelty in population-based search itself.
>
> Instead, integrating BehaveSim into the LLM-AAD framework is one of its use cases. While the LLM-AAD paradigm can be improved along many dimensions (e.g., better prompting, improved interaction mechanisms with LLMs, or population-based search strategies), a central challenge in any diversity-driven approach is defining how to measure diversity.
>
> Here, our contribution is to introduce a behavioral perspective into LLM-AAD: through a niche-based mechanism, we incorporate BehaveSim into *FunSearch* to explicitly encourage diversity in problem-solving behavior during the search process, demonstrating that encouraging diversity in problem-solving behavior leads to improved AAD performance.

---

### Official Review · Reviewer_rhhD · 2025-11-01

**Soundness:** 2
**Presentation:** 2
**Contribution:** 2
**Rating:** 2
**Confidence:** 2

**Summary:**

This paper introduces BehaveSim, a novel similarity metric for measuring algorithm similarity from a behavioral perspective rather than code structure.The paper demonstrates that existing code similarity metrics (token-based, AST-based, embedding-based, execution-based) fail to capture algorithmic behavioral differences. To solve this problem, the paper proposes measuring similarity via problem-solving trajectories - sequences of intermediate solutions generated during algorithm execution, compared using Dynamic Time Warping (DTW).

**Strengths:**

The curated dataset with 4 algorithm pair types (varying code/behavior/result similarity combinations) provides rigorous validation. The results clearly demonstrate that BehaveSim achieves 1.0 similarity on Type-3 pairs (same behavior, different code) while existing code metrics fail, and correctly identifies behavioral differences where code metrics show high similarity.

**Weaknesses:**

1. The evaluation methodology does not use any AI models or AI-related methods. BehaveSim is essentially a general algorithm comparison technique based on execution traces and DTW, which appears equally applicable to comparing human-written code. The source of code (LLM-generated versus human-written) seems irrelevant to the core methodology, raising questions about whether this is fundamentally a software engineering contribution rather than an AI/ML contribution suitable for ICLR.
2. The method only applies to iterative algorithms such as sorting, search, and optimization, excluding ML algorithms and non-iterative approaches. This significantly limits the practical applicability and generalizability of the approach.
3. The similarity dataset contains only two AAD tasks, and all code exampels are hand-crafted, well-known algorithms. The experimental scope is very limited. Besides, the paper does not demonstrate whether BehaveSim can distinguish novel, unseen LLM-generated algorithms, which is critical for the claimed contribution to "algorithm discovery." This represents a significant gap between the evaluation (known algorithms) and the application (discovering novel algorithms).

**Questions:**

See weakness.

---

> ### Author Response · Authors · 2025-11-21
> **Response to Reviewer rhhD**
>
> > **Weakness 1:** The evaluation methodology does not use any AI models or AI-related methods. BehaveSim is essentially a general algorithm comparison technique based on execution traces and DTW, which appears equally applicable to comparing human-written code. The source of code (LLM-generated versus human-written) seems irrelevant to the core methodology, raising questions about whether this is fundamentally a software engineering contribution rather than an AI/ML contribution suitable for ICLR.
>
> We respectfully disagree with the reviewer's statement, which we believe is factually incorrect. Our work is positioned within the rapidly growing area of AI (LLM)-based Automated Algorithm Design (LLM-AAD) (References 1-6).
>
> Beyond proposing new algorithm-generation methods, we believe it is equally important for this field to address a central question: How to measure the similarity of algorithms' problem-solving behavior.
>
> This basic issue is critically important in the context of LLM-AAD, where LLMs produce a vast amount of algorithms whose problem-solving patterns are embedded in the code. Distinguishing algorithms based on their problem-solving behavior can help LLM-AAD methods identify potentially novel algorithms that exhibit different problem-solving behaviors rather than merely better performance.
>
> Although BehaveSim itself does not rely on AI models, our study directly contributes to AI-based methods in two important ways: (i) we compare BehaveSim against several AI-driven code similarity methods to show that they fail to capture behavioral similarity; and (ii) we empirically demonstrate that incorporating BehaveSim into state-of-the-art LLM-AAD methods leads to consistent and significant improvements in search performance. In other words, BehaveSim can enhance the capability of existing AI-based methods to design better algorithms.
>
> For these reasons, we believe our paper is within the scope of ICLR and of interest to general AI researchers.
>
> **References:**
>
> 1. Romera-Paredes et. al. Mathematical Discoveries from Program Search with Large Language Models. Nature. 2024.
> 2. Ma et. al. Eureka: Human-level Reward Design via Coding Large Language Models. ICLR 2024.
> 3. Shojaee et. al. LLM-SR: Scientific Equation Discovery via Programming with Large Language Models. ICLR 2025.
> 4. Wang et. al. Planning in Natural Language Improves  LLM Search for Code Generation. ICLR 2025.
> 5. Liu et. al. Evolution of heuristics: Towards Efficient Automatic Algorithm Design Using Large Language Model. ICML 2024.
> 6. Zheng et. al. Monte Carlo Tree Search for Comprehensive Exploration in LLM-Based Automatic Heuristic Design. ICML 2025.
>
>
>
> > **Weakness 2:** The method only applies to iterative algorithms such as sorting, search, and optimization, excluding ML algorithms and non-iterative approaches. This significantly limits the practical applicability and generalizability of the approach.
>
> ### For non-iterative algorithms
>
> When an algorithm does not produce a multi-step problem-solving trajectory, the trajectory degenerates into a single result solution. In such cases, BehaveSim becomes equivalent to execution result-based comparison.
>
> ### Scope of iterative algorithms
>
> We acknowledge that the imprecise wording in our scope of "iterative algorithm" may have caused confusion for the reviewer. We have revised the description to more accurately reflect the intended meaning.
>
> We have revised the description to clarify that the scope of "iterative algorithm" encompasses methods that progressively refine intermediate solutions, rather than solving analytically in a single step. Formally, we express this as $x_{t+1} = f(x_t)$, where $x_t$ is the solution (or partial solution) at step $t$, and $f$ is the iterative algorithm mapping $x_t$ to $x_{t+1}$.
>
> ### BehaveSim for ML algorithms
>
> Many ML algorithms fall naturally within this scope. For instance, in *K-Means*, $x_t$ represents cluster assignments and centroids updated by $f$. Similarly, in Reinforcement Learning (RL), $x_t$ corresponds to policy parameters or agent states evolving via environment interactions. Both examples generate solution trajectories fully compatible with BehaveSim.

---

> ### Author Response · Authors · 2025-11-21
> **Response to Reviewer rhhD**
>
> > **Weakness 3:** The similarity dataset contains only two AAD tasks, and all code examples are hand-crafted, well-known algorithms. The experimental scope is very limited. Besides, the paper does not demonstrate whether BehaveSim can distinguish novel, unseen LLM-generated algorithms, which is critical for the claimed contribution to "algorithm discovery." This represents a significant gap between the evaluation (known algorithms) and the application (discovering novel algorithms).
>
> We thank the reviewer for raising these concerns. We believe the reviewer may misunderstand the similarity dataset.
>
> ### Clarification for the similarity dataset and its scope
>
> Our dataset contains 30 algorithm pairs, including matrix multiplication, classical algorithms (e.g., searching and sorting), and hand-crafted heuristics. However, the dataset does not include any AAD tasks.
>
> The purpose of this similarity dataset is to empirically validate our claim that existing code similarity metrics are insufficient for capturing behavioral similarity for algorithms. To this end, we select classical algorithms whose problem-solving behavior is well understood and provides a clear ground truth.
>
> From this perspective, although the benchmark is based on classical examples, it is sufficient for testing our hypothesis. As shown in Table 2 in our manuscript, none of the tested baselines can reliably distinguish these well-studied algorithms despite their substantial differences in problem-solving behavior. We believe this already provides strong evidence to support our claim.
>
> ### On distinguishing novel, unseen LLM-generated algorithms
>
> We thank the reviewer for the insightful comment. Based on this feedback, we have refined the description of our contribution to more precisely reflect the intended scope.
>
> BehaveSim is designed to differentiate algorithms from the perspective of problem-solving behavior, which is particularly significant for LLM-AAD, a promising new paradigm for future automated algorithm design.
>
> By utilizing BehaveSim to explicitly encourage behavioral diversity in LLM-AAD, we theoretically enable the search to explore a wider range of distinct problem-solving strategies. In this context, BehaveSim may serve as a useful component in facilitating the discovery of novel algorithms.

---

> ### Comment · Reviewer_rhhD · 2025-11-25
> **Official Comment by Reviewer rhhD**
>
> Thank you for the rebuttal and additional experiments. I appreciate the significant effort to integrate BehaveSim with LLM-based evolving algorithms, which addresses my main concern. However, due to the limited rebuttal period and the issues with the insufficient original experimental design, the new experiments only cover three tasks, which is insufficient to fully validate the approach's effectiveness in automatic algorithm discovery.
> I believe this work has merit and would benefit from a revision with more comprehensive experiments before submission. Based on this, I am raising my score to 4.

---

> > ### Author Response · Authors · 2025-11-26
> > **Response to Reviewer rhhD**
> >
> > We sincerely thank the reviewer for the insightful comments and for acknowledging the merit of our work. We appreciate knowing that we have successfully addressed the reviewer's main concern.
> >
> > ### Summary of experiments and their intended goals
> >
> > To demonstrate that the experiments sufficiently support our hypothesis, we summarize the experiments included in the manuscript and their intended goals. Experiments present in the original submission are indicated in **bold**.
> >
> > 1. *Benchmarking the limitations of existing code similarity methods*.
> >    To investigate whether existing similarity methods can measure behavioral similarity, we construct a dataset comprising classic algorithms and handcrafted heuristics, for which their behavior is well understood. We test 12 methods across six categories on the dataset.
> >
> >    - **The original submission includes eight methods spanning five methodological categories.** In rebuttal, we further add four methods: one embedding model (Qwen3-Embedding-0.6B) and three execution-trace-based methods.
> >    - Results are presented in Table 2, which indicates that these methods do not reliably capture behavioral similarity.
> >
> > 2. *Ablation studies for BehaveSim.*
> >    We conduct ablation studies on trajectory distance metrics and hyperparameters governing trajectory truncation and sampling.
> >
> >    - **We include both visual and empirical analysis on trajectory distance choices.** We perform ablation studies on the hyperparameters of trajectory truncating and sampling.
> >    - The results are presented in Figures 3, 6, 7, and Table 9. These experiments demonstrate how these hyperparameters influence the BehaveSim.
> >
> > 3. *Use cases of BehaveSim in automated algorithm design (AAD).*
> >    Having established BehaveSim's ability to capture behavioral similarity, we evaluate whether incorporating BehaveSim into LLM-AAD methods improves performance. We integrate BehaveSim into two existing methods:
> >
> >    - **(i) *FunSearch+BehaveSim***, and (ii) *EoH+BehaveSim*
> >
> >    We evaluate these enhanced methods on three challenging AAD tasks:
> >
> >    - **(i) Admissible Set Problem (ASP)**, **(ii)Traveling Salesman Problem (TSP)**, and (iii) Circle Packing Problem
> >    - The results presented in Figure 4 and Table 3 demonstrate that encouraging behavioral diversity significantly improves the performance of both *FunSearch* and *EoH*.
> >
> >    We also incorporate analysis for *FunSearch+BehaveSim*:
> >
> >    - We perform an **ablation study on the inter-island selection probability $p_{s1}$**, and **visualize inter- and intra-island behavioral diversity over the course of evolution.**
> >    - The results of the ablation study are demonstrated in Figure 10. The visualization is shown in Figure 9.
> >
> > 4. *Use case of BehaveSim in algorithm analysis.*
> >
> >    - **We analyze the discovered algorithms using BehaveSim and CodeBLEU.**
> >    - Results in Figure 5 demonstrate that BehaveSim can identify algorithms that exhibit distinct problem-solving behaviors despite having similar code, as well as the opposite cases, showing that BehaveSim can characterize algorithms from a behavioral perspective rather than relying solely on code-level similarity.
> >
> > ### Regarding the reviewer's comment on the insufficient original experimental design
> >
> > We agree that the original experimental evaluations could be expanded; however, we respectfully assert that they are sufficient to support our claims. As summarized above, in our original submission, each claim is supported by curated experiments, ranging from comparisons on classical algorithm pairs, to method-level ablations, to integration within AAD. Therefore, we believe our original experimental design is sufficient.
> >
> > ### Regarding the reviewer's comment on insufficient validation for automatic algorithm discovery effectiveness
> >
> > We would like to clarify that the primary contribution of the paper is not proposing a new AAD method. Instead, the primary contribution of this work is to introduce a tangible method for measuring behavioral similarity for algorithms.
> >
> > Using BehaveSim to enhance AAD performance is presented as one practical use case. We believe the scope and scale of our AAD experiments are comparable to existing AAD papers, for example, EoH [1] and HSEvo [2] evaluate three AAD tasks in the main text.
> >
> > With one week remaining in the discussion period, if the reviewer has specific suggestions for additional tasks, we will make every effort for addressing the reviewer's concerns.
> >
> > **References:**
> >
> > 1. Liu et al. Evolution of Heuristics: Towards Efficient Automatic Algorithm Design Using Large Language Models. ICML 2024.
> > 1. HSEvo: Elevating Automatic Heuristic Design with Diversity-Driven Harmony Search and Genetic Algorithm Using LLMs. AAAI 2025.

---

### Official Review · Reviewer_SVva · 2025-11-03

**Soundness:** 3
**Presentation:** 2
**Contribution:** 3
**Rating:** 6
**Confidence:** 3

**Summary:**

This paper proposes BehaveSim, a new metric for measuring algorithm similarity from a behavioral perspective rather than focusing on code structure or final outputs. The key idea is to represent an algorithm by its problem-solving trajectory, which records intermediate solutions generated step by step. The similarity between two algorithms is then defined as the resemblance between their trajectories, measured through DTW. The authors demonstrate that BehaveSim can distinguish algorithms that are behaviorally different but structurally similar, and that integrating this measure into LLM-AAD improves search performance by promoting behavioral diversity.

**Strengths:**

[1] The paper addresses an important and overlooked gap by redefining algorithm similarity from the behavioral viewpoint. This is a clear and original contribution.

[2] The distinction among code-level, behavior-level, and result-level similarity is well presented, and the taxonomy of four types of algorithm pairs is intuitive and pedagogically useful.

[3] The idea of representing algorithm behavior through trajectories and computing DTW-based similarity is theoretically grounded and applicable to both continuous and discrete problems.

**Weaknesses:**

[1] BehaveSim is currently designed for iterative algorithms only. It cannot yet handle recursive, dynamic programming, or machine-learning-based algorithms. This limits its generality.

[2] Several heuristic parameters, such as trajectory truncation, normalization constants, and distance scaling, are not systematically analyzed. Their influence on stability and reproducibility is unclear.

[3] The benchmark for similarity evaluation mainly includes synthetic or classical algorithm examples. Broader testing on more diverse algorithmic domains would strengthen general claims.

[4] The integration details of BehaveSim into the multi-island FunSearch framework are brief in the main text, making reproduction difficult without reading the appendix.

[5] Some recent semantic or execution-trace-based code similarity methods are discussed only briefly without quantitative comparison.

**Questions:**

[1] Could the authors elaborate on how BehaveSim would handle non-iterative algorithms or those with stochastic internal states (e.g., randomized search or Monte Carlo methods)?

[2] Would the DTW-based comparison still be meaningful in these contexts, and if not, how might the behavioral similarity concept be adapted?

---

> ### Author Response · Authors · 2025-11-21
> **Response to Reviewer SVva**
>
> > **Weakness 1:** BehaveSim is currently designed for iterative algorithms only. It cannot yet handle recursive, dynamic programming, or machine-learning-based algorithms. This limits its generality.
>
> We acknowledge that the imprecise wording in our scope of "iterative algorithm" may have caused confusion for the reviewer. We have revised the description to more accurately reflect the intended meaning.
>
> ### Scope of iterative algorithms
>
> We clarify that an "iterative algorithm" here refers to methods producing a sequence of progressively refined intermediate solutions, formally $x_{t+1} = f(x_t)$, where $x_t$ is the solution (or state) at step $t$ and $f$ is the iterative algorithm. This scope encompasses:
>
> - **Recursive algorithms.** Recursive procedures fit this scope by treating recursive calls as state transitions. For instance, the recursive DFS in Figure 1(a) in the manuscript updates the traversal route (partial solution) at each step, forming a problem-solving trajectory trackable by BehaveSim.
> - **Dynamic programming (DP).** The state $x_t$ can be defined as the DP table after computing $t$ subproblems. The transition $f$ applies the recurrence relation to generate the next entry, producing a sequence of table updates.
> - **Machine learning algorithms.** Many ML methods are inherently iterative. For example, in K-Means, $x_t$ represents cluster assignments; in Reinforcement Learning (RL), $x_t$ denotes policy parameters or agent states. Both evolve through iterative updates compatible with BehaveSim's formulation.
>
>
>
>
> > **Weakness 2:** Several heuristic parameters, such as trajectory truncation, normalization constants, and distance scaling, are not systematically analyzed. Their influence on stability and reproducibility is unclear.
>
> We thank the reviewer for this suggestion. We have conducted additional analyses to clarify the impact of these parameters.
>
> ### Trajectory truncation parameters
>
> BehaveSim controls trajectory granularity via (i) *Truncation* (removing the last proportion $k$ of the trajectory) and (ii) *Sampling* (keeping one solution every $n$ steps). We analyze the sensitivity of similarity scores to these parameters, with results shown in Table 1 below.
>
> We observe that the metric is robust to light-to-moderate processing. Truncating 0.1-0.3 of the trajectory yields scores nearly identical to the full trajectory, and deviations only become noticeable at aggressive ratios (0.4-0.5). Similarly, sampling intervals of 1-2 steps have minimal effect, while larger intervals (3-5) introduce expected variance. This confirms that BehaveSim is stable under reasonable parameter choices.
>
>
>
> **Table 1:** Influence of trajectory sampling strategies on similarity scores.
>
> | **Method Name**            | **Type-1** | **Type-2** | **Type-3** | **Type-4** |
> | -------------------------- | ---------- | ---------- | ---------- | ---------- |
> | Full-Trajectory (k=0, n=0) | 0.56       | 0.73       | 1.00       | 0.46       |
> | k=0.1, n=0                 | 0.55       | 0.73       | 1.00       | 0.47       |
> | k=0.2, n=0                 | 0.55       | 0.73       | 1.00       | 0.48       |
> | k=0.3, n=0                 | 0.57       | 0.73       | 1.00       | 0.52       |
> | k=0.4, n=0                 | 0.60       | 0.73       | 1.00       | 0.58       |
> | k=0.5, n=0                 | 0.65       | 0.75       | 1.00       | 0.64       |
> | k=0, n=1                   | 0.57       | 0.72       | 1.00       | 0.52       |
> | k=0, n=2                   | 0.56       | 0.74       | 1.00       | 0.59       |
> | k=0, n=3                   | 0.56       | 0.72       | 1.00       | 0.65       |
> | k=0, n=4                   | 0.56       | 0.82       | 1.00       | 0.72       |
> | k=0, n=5                   | 0.62       | 0.82       | 1.00       | 0.74       |
>
>
>
> ### Normalization constants
>
> We clarify that normalization constants are *deterministic, problem-specific values*, not tunable heuristics. Their purpose is to scale scores to a comparable $[0,1]$ range (to combine with other distance metrics when needed).
>
> - For *categorical/permutation* solutions, we normalize by the theoretical maximum edit distance.
> - For *continuous/discrete* solutions, we use the domain's upper bound constraints. For problems without such bound constraints, we normalize them using the maximum observed pairwise distance. While in this case, this does not strictly guarantee the $[0,1]$ range for all cases, in practice, the values typically fall within it.
>
> Since these values are strictly derived from the problem definition, they do not introduce randomness or reproducibility issues.

---

> ### Author Response · Authors · 2025-11-21
> **Response to Reviewer SVva**
>
> > **Weakness 3:** The benchmark for similarity evaluation mainly includes synthetic or classical algorithm examples. Broader testing on more diverse algorithmic domains would strengthen general claims.
>
> We appreciate the reviewer's comments and agree that our benchmark primarily consists of synthetic or classical algorithms.
>
> Our general claim is that existing code similarity metrics are insufficient for capturing behavioral similarity for algorithms. To this end, we select classical algorithms whose problem-solving behavior is well understood and provides a clear ground truth.
>
> From this perspective, although the benchmark is based on classical examples, it is sufficient for validating our hypothesis. As shown in Table 2 in our manuscript, none of the tested baselines can reliably distinguish these well-studied algorithms despite their substantial differences in problem-solving behavior. We believe this already provides strong evidence to support our claim.
>
>
>
>
>
> > **Weakness 4:** The integration details of BehaveSim into the multi-island FunSearch framework are brief in the main text, making reproduction difficult without reading the appendix.
>
> We thank the reviewer for pointing out that the integration details of BehaveSim within the multi-island *FunSearch* were insufficiently detailed in the main text.
>
> The core of *FunSearch+BehaveSim* is to encourage behavioral diversity by organizing algorithms into behaviorally cohesive islands. The process proceeds as follows:
>
> 1. Initialization: The database is initialized by generating algorithms and clustering them into islands based on BehaveSim similarity.
> 2. Selection: The LLM generates new candidates using few-shot examples selected via inter- or intra-island sampling strategies.
> 3. Database management: New candidates are registered on the island with the most similar behavior, maintaining local coherence while ensuring global behavioral diversity.
>
> We will incorporate the overall procedures for *FunSearch+BehaveSim* in the manuscript to facilitate a clearer understanding.

---

> ### Author Response · Authors · 2025-11-21
> **Response to Reviewer SVva**
>
> > **Weakness 5:** Some recent semantic or execution-trace-based code similarity methods are discussed only briefly without quantitative comparison.
>
> We thank the reviewer for pointing out the incompleteness of our comparison. We conduct additional analyses below.
>
> ### Comparison to execution-trace-based methods
>
> From a methodological perspective, existing execution-trace-based approaches [1-3] fundamentally differ from BehaveSim in that they track low-level internal states, such as variable values, instruction-level changes, or memory attributes. In contrast, the problem-solving trajectory we use can be viewed as a highly specialized trace, corresponding to tracking the variation of a solution or partial solution in the algorithm.
>
> For general-purpose program analysis, traces for multiple variables and function calls indeed contain rich and useful information. However, for characterizing problem-solving behavior, the large amount of low-level state changes (multiple variable states, function-call logs, etc.) may introduce excessive information unrelated to the algorithm's problem-solving dynamics, which could potentially obscure similarity calculations.
>
> Empirically, we re-implemented NeXT [1] using `pysnooper` [4] to trace internal variable dynamics. Since NeXT does not define a similarity metric, we converted the traces into tokens and embeddings, measuring similarity via N-Gram-based methods (e.g., BLEU) and embedding models (Jina-Code-Embedding, Qwen3-Embedding-0.6B).
>
> As shown in Table 2 below, these methods exhibit high similarity scores for dissimilar behaviors (Types 1 and 2), indicating that they fail to distinguish behavioral differences. This confirms that measuring similarity via raw execution traces is less effective than BehaveSim's problem-solving trajectory, which comprises intermediate solutions.
>
> ### Comparison to semantic-based methods
>
> Our manuscript already included comparisons with semantic embeddings, such as *CodeBERTScore* and *Jina-Code-Embedding* (this model is referred to as "CodeEmbedding" in our manuscript). To further strengthen this part, we additionally evaluated a semantic embedding model, *Qwen3-Embedding-0.6B*. As shown in Table 2 below, we consistently observed that these models fail to capture behavioral similarity.
>
> These additional results reinforce our main claim: neither semantic embeddings nor low-level execution traces reliably reflect algorithm behavior. BehaveSim's explicit focus on the dynamics of problem-solving solutions makes it a more suitable method for this purpose.
>
>
>
> **Table 2:** Average similarity on four types of data calculated by various metrics. Methods labeled with "*" are newly added experiments.
>
> | **Method Type**                     | **Method Name**          | **Type-1** | **Type-2** | **Type-3** | **Type-4** |
> | ----------------------------------- | ------------------------ | ---------- | ---------- | ---------- | ---------- |
> | Based on Token Match                | *ROUGE*                  | 0.95       | 0.96       | 0.70       | 0.47       |
> |                                     | *BLEU*                   | 0.83       | 0.94       | 0.42       | 0.16       |
> |                                     | *CrystalBLEU*            | 0.97       | 0.99       | 0.68       | 0.51       |
> | Based on Structure                  | *AST*                    | 0.96       | 1.00       | 0.76       | 0.57       |
> | Combine Token Match and Structure   | *CodeBLEU*               | 0.97       | 0.94       | 0.91       | 0.75       |
> | Based on Embedding                  | *CodeBertScore*          | 0.84       | 0.97       | 0.60       | 0.38       |
> |                                     | *Jina-Code-Embedding*    | 0.99       | 0.99       | 0.90       | 0.84       |
> |                                     | *Qwen3-Embedding-0.6B*\* | 0.94       | 0.93       | 0.87       | 0.73       |
> | Based on Execution Results          | --                       | 1.00       | 0.00       | 1.00       | 1.00       |
> | Based on Execution Trace            | *BLEU Similarity*\*      | 0.86       | 0.95       | 0.61       | 0.54       |
> |                                     | *Jina-Code-Embedding*\*  | 1.00       | 1.00       | 0.87       | 0.77       |
> |                                     | *Qwen3-Embedding-0.6B*\* | 0.99       | 1.00       | 0.91       | 0.78       |
> | Similarity of their behavior (Ours) | *BehaveSim*              | 0.56       | 0.73       | 1.00       | 0.46       |
>
>
>
> **References:**
>
> 1. Ni et. al. NExT: Teaching Large Language Models to Reason about Code Execution. ICML 2024.
> 2. Pei et. al. TREX: Learning Execution Semantics from Micro-Traces for Binary Similarity. IEEE Transactions on Software Engineering. 2023.
> 3. Wang et. al. Combining Structured Static Code Information and Dynamic Symbolic Traces for Software Vulnerability Prediction. ICSE 2024.
> 4. Rachum et. al. PySnooper: Never use print for debugging again. URL https://github.com/cool-RR/PySnooper. 2019.

---

> ### Author Response · Authors · 2025-11-21
> **Response to Reviewer SVva**
>
> > **Question 1:** Could the authors elaborate on how BehaveSim would handle non-iterative algorithms or those with stochastic internal states (e.g., randomized search or Monte Carlo methods)?
>
> > **Question 2:** Would the DTW-based comparison still be meaningful in these contexts, and if not, how might the behavioral similarity concept be adapted?
>
> ### For non-iterative algorithms
>
> When an algorithm does not produce a multi-step problem-solving trajectory, the trajectory degenerates into a single result solution. In such cases, BehaveSim becomes equivalent to execution result-based comparison, and DTW no longer plays a meaningful role because there is no temporal structure to align.
>
> ### For algorithms with stochastic internal states
>
> Stochastic intermediate states do not invalidate the concept of behavioral similarity, but we require a new way to measure the pairwise distance.
>
> 1. **Sample multiple solutions at each step.** If the transition $x_{t+1}=f_A(x_t \mid \xi)$ involves a random noise $\xi$, we obtain multiple samples to approximate the distribution $P_{t+1}(x)$ and similarly obtain $P_{t+1}(y)$ for algorithm $B$. The pairwise distance can then be defined as a divergence $D(P_{t+1}(x), P_{t+1}(y))$, and DTW can be applied over these distributional distances. This preserves temporal alignment while accounting for stochasticity.
> 2. **Fixing random seed.** Fixing the random seed is another way to reduce the randomness. In this case, DTW can then be applied directly.
>
> Both methods will further aggregate similarity across multiple problem instances (as mentioned in Sec. 3, lines 353-360 in our revised manuscript), which naturally reduces variance induced by randomness.
>
> Sampling multiple solutions at each step is more robust since it compares the behavioral distributions, but it requires more computational overhead for sampling. Fixing random seeds ensures reproducibility, but may lead to biased calculations of behavioral similarity. Depending on the specific context, either approach can be selected accordingly.

---

### Author Response · Authors · 2025-11-24
**General Response: Summary of Manuscript Revisions**

We sincerely thank all reviewers for their constructive comments and the time dedicated to reviewing our paper. We have carefully considered all suggestions and revised the manuscript to address the concerns raised. The major updates in the revised manuscript are summarized as follows:

### 1. Refining the wording of the scope of the iterative algorithm

**(Reviewer SVva, Weakness 1; Reviewer rhhD, Weakness 2; Reviewer KsNQ, Weakness 1; Reviewer TTPL, Weakness 1)**

In Section 3, we have revised our statement regarding the scope of the iterative algorithms. We have clarified that our scope is intended to be broad, inclusive of recursive algorithms, machine learning models, etc., as highlighted by the reviewers.

### 2. Experiments and discussions on execution-trace-based methods

**(Reviewer SVva, Weakness 5; Reviewer KsNQ, Weakness 4)**

- In Section 2.1 (Revisiting Code Similarity), we included a discussion on execution-trace-based methods.
- Furthermore, we added comparative experiments and detailed analysis of these methods in Section 2.2.

### 3. Elaborating on the BehaveSim method

- **(Reviewer SVva, Weakness 2):** In Section 3, we added specific details regarding trajectory truncation and sampling. We also provided experimental analysis for these choices in Appendix C.
- **(Reviewer SVva, Question 1; Reviewer KsNQ, Weakness 1):** In Section 3, we explicitly clarified that BehaveSim computes similarity by averaging across multiple problem instances and starting points. We also detailed the solution for handling algorithms with stochastic intermediate states.

### 4. Additional experiments and implementation details

- **(Reviewer SVva, Weakness 4):** In Section 4.1, we added a detailed description of the FunSearch+BehaveSim implementation.
- **(Reviewer KsNQ, Weakness 5):** In Section 4.1, we extended our evaluation to include the Circle Packing Problem (CPP) and introduced a new combination of BehaveSim, *EoH+BehaveSim*. The results are shown in Table 3 and Figure 4 in the revised version.

---

### Author Response · Authors · 2025-12-03
**Authors' Summary to Area Chair (Part 1/2)**

We sincerely thank AC and the reviewers for the time and effort devoted to evaluating our work. We are encouraged that the reviewers recognize the novelty and importance of redefining algorithm similarity from a behavioral perspective, as well as the effectiveness of BehaveSim in distinguishing algorithms that are behaviorally different but structurally similar.

To support the AC's final assessment, we provide a concise summary of our paper along with the review comments.

## 1. Recap of Motivations and Contributions

**Motivations.** With the rapid development of large language models (LLMs), an increasing number of algorithms are now AI-assisted or AI-generated. This trend yields an urgent need for automated methods to evaluate whether two algorithms are inherently distinct or merely syntactic refinements of an existing one. Since algorithms are primarily represented as code, measuring similarity at the code level is a natural choice.

However, existing code similarity techniques are primarily designed to compare programs through static features such as tokens, structure, or embeddings. These approaches work well for general software analysis but fall short for algorithms, whose essence lies in the dynamic behavior they exhibit during problem solving rather than their static code. As a result, static code similarity metrics, e.g., AST similarity, CodeBLEU, and Jina-Code-Embedding, are insufficient for capturing the behavioral characteristics that truly distinguish one algorithm from another.

**Contributions.** This work proposes BehaveSim, a tangible method for measuring behavioral similarity for algorithms. The problem-solving behavior of an algorithm is represented as a trajectory consisting of intermediate solutions, and BehaveSim is computed by comparing the similarity of such trajectories. We verify the effectiveness of BehaveSim on a curated dataset (Sec. 2.1).

We further demonstrate two use cases of BehaveSim:

1. We show that encouraging behavioral diversity in Language Model-based Automated Algorithm Design (LLM-AAD) methods significantly improves their performance. We integrate BehaveSim into two existing methods: (i) FunSearch+BehaveSim and (ii) EoH+BehaveSim, and demonstrate their effectiveness on three challenging AAD tasks: (i) the Admissible Set Problem (ASP), (ii) the Traveling Salesman Problem (TSP), and (iii) the Circle Packing Problem (CPP).
2. We show that BehaveSim can identify algorithms that exhibit distinct problem-solving behaviors despite having similar code, as well as the converse.

---

> ### Author Response · Authors · 2025-12-03
> **Authors' Summary to Area Chair (Part 2/2)**
>
> ## 2. Summary of Review Comments
>
> Overall, the reviewers provided positive feedback on the paper's novelty, rigorous benchmark, and principled methodology. Reviewers **SVva**, **KsNQ**, and **TTPL** agreed that the work identifies an overlooked gap in algorithm evaluation by shifting the focus from code-level or output-level metrics to a behavioral perspective, with Reviewer **SVva** describing this as a clear and original contribution, and Reviewers **TTPL** and **KsNQ** emphasizing its novelty and strong motivation. Reviewers **SVva**, **rhhD**, and **KsNQ** also commended the rigor of our benchmark design, noting that the taxonomy of algorithm-pair types and the curated dataset provide convincing empirical evidence.
>
> In addition, Reviewers **SVva** and **KsNQ** appreciated the principled formulation of the method, particularly its concrete representation of algorithm behavior through trajectories, with Reviewer **KsNQ** remarking that the methodology is well-formalized, reproducible, and extensible.
>
> ### 2.1. Scope of the Iterative Algorithm
>
> Reviewers **(SVva, W1; rhhD, W2; KsNQ, W1; TTPL, W1)** raised concerns about the limited scope of "iterative algorithms" in BehaveSim.
>
> We clarify that an "iterative algorithm" in this work refers to any method that produces a sequence of progressively refined intermediate solutions, formally $x_{t+1} = f(x_t)$, where $x_t$ is the solution (or partial solution) at step $t$ and $f$ denotes the algorithmic update. This scope encompasses recursive algorithms, machine learning algorithms, dynamic programming algorithms, and differentiable algorithms, as mentioned by the reviewers.
>
> Algorithms with stochastic internal states also fall within the scope of BehaveSim. We propose two approaches for handling randomness: (i) sampling multiple trajectories at each step, and (ii) fixing the random seed.
>
> We have revised Sec. 3 of the manuscript to clarify the scope of iterative algorithms and our solutions for handling algorithms with stochastic states.
>
> ### 2.2. Baselines for BehaveSim
>
> Reviewers **(SVva, W5; KsNQ, W4)** suggested discussing and comparing BehaveSim with execution-trace-based methods on the curated similarity dataset.
>
> Execution trace-based methods measure similarity between low-level program states, such as variable values at executed lines, fine-grained instruction-level changes (e.g., x86 instructions), or variable attributes (e.g., addresses, sizes). However, for characterizing problem-solving behavior, these extensive low-level changes introduce a large amount of information that is unrelated to problem-solving dynamics, potentially obscuring the computaion of problem-solving similarity.
>
> To verify whether execution-based methods can capture behavioral similarity of algorithms, we have empirically evaluated these methods on the similarity dataset in Sec. 2 of the manuscript. The results demonstrate that execution-trace-based methods are also incapable of measuring behavioral similarity.
>
> ### 2.3. Scope of Experiments
>
> Reviewers **(SVva, W3; rhhD, W3)** noted that the similarity dataset is limited to synthetic or classical algorithm examples.
>
> We argue that the purpose of this dataset is to empirically validate our claim that existing code similarity metrics are insufficient for capturing behavioral similarity. Classical algorithms provide a well-understood behavioral ground truth, making them ideal for evaluating this specific hypothesis.
>
> From this perspective, although the benchmark is based on classic examples, it is sufficient to support our claim. As shown in Table 2 of the manuscript, none of the evaluated baselines can reliably distinguish these well-studied algorithms despite their substantial behavioral differences, providing strong evidence in support of our hypothesis.
>
> Reviewer **(KsNQ, W5)** further suggested that evaluating the integration of BehaveSim with LLM-AAD methods on broader and more complex AAD tasks would further strengthen the contribution.
>
> To address this, we additionally integrate BehaveSim into EoH and evaluate both *FunSearch+BehaveSim* and *EoH+BehaveSim* on a new challenging benchmark: the Circle Packing Problem (CPP).
>
> The results, now included in Sec. 4.1, show that both *FunSearch+BehaveSim* and *EoH+BehaveSim* consistently outperform their original counterparts on CPP, indicating that encouraging behavioral diversity in LLM-AAD leads to consistent and meaningful performance improvements.

---

### Meta-Review · Area_Chair_iJDp · 2025-12-10

**Summary:**

Thank you to the reviewers for your valuable suggestions from multiple perspectives. Overall, I think their main problems at present lie in:



- Lack of generality.

- Novelty is overstated.

- Lack of theoretical support.

- The experimental results are not sufficient and need to be further supplemented.



In addition, some reviewers mentioned problems such as the matching between the paper and ICLR.

**Reviewer Concerns:**

I am very grateful for the reply provided by the author. I believe that some of the reviewers' questions will be resolved, such as some supplements regarding the experimental results.

However, the doubts of some reviewers about the generality of the methods may not be addressed because more empirical evidence cannot be provided.

Overall, the explanation regarding universality still needs further supplementation. Considering that the current attitude of the reviewers tends to be positive (although the confidence level is very low), I tend to accept this paper.

**Reviewer Scores:**

For Reviewer SVva, he may accept most of the explanations. I think the reviewer will maintain the current score (**Rating:** 6).



For Reviewer rhhD, his discussion made his point clear. I think this reviewer will increase the current score (**Rating:** 2 to 4).



For Reviewer KsNQ, he may accept most of the explanation. I think the reviewer will maintain the current score (**Rating:** 6).



For Reviewer TTPL, his discussion made his point clear. I think the reviewer will maintain the current score (**Rating:** 6).

---

### Decision · Program_Chairs · 2026-01-26

Accept (Poster)